# PROTEINBENCH: A HOLISTIC EVALUATION OF PROTEIN FOUNDATION MODELS

**Fei Ye**[*], **Zaixiang Zheng**[*], **Dongyu Xue**[*], **Yuning Shen**[*], **Lihao Wang**[*]
**Yiming Ma, Yan Wang, Xinyou Wang, Xiangxin Zhou** and **Quanquan Gu**[†]
ByteDance Research
{yefei.joyce,quanquan.gu}@bytedance.com

Project page: https://proteinbench.github.io/

## ABSTRACT

Recent years have witnessed a surge in the development of protein foundation models, significantly improving performance in protein prediction and generative tasks ranging from 3D structure prediction and protein design to conformational dynamics. However, the capabilities and limitations associated with these models remain poorly understood due to the absence of a unified evaluation framework. To fill this gap, we introduce ProteinBench, a holistic evaluation framework designed to enhance the transparency of protein foundation models. Our approach consists of three key components: (i) A taxonomic classification of tasks that broadly encompass the main challenges in the protein domain, based on the relationships between different protein modalities; (ii) A multi-metric evaluation approach that assesses performance across four key dimensions: quality, novelty, diversity, and robustness; and (iii) In-depth analyses from various user objectives, providing a holistic view of model performance. Our comprehensive evaluation of protein foundation models reveals several key findings that shed light on their current capabilities and limitations. To promote transparency and facilitate further research, we release the evaluation dataset, code, and a leaderboard publicly for further analysis and a general modular toolkit. We intend for ProteinBench to be a living benchmark for establishing a standardized, in-depth evaluation framework for protein foundation models, driving their development and application while fostering collaboration within the field.

## 1 INTRODUCTION

Proteins are fundamental molecules playing pivotal roles in a vast array of biological processes, from enzymatic catalysis and signal transduction to structural support and immune response. Their functions are determined by their amino acid sequences, often mediated through folding into specific three-dimensional structures. Understanding the complex interplay between protein sequence, structure, and function is crucial for advancing science and engineering spanning pharmaceuticals, agriculture, specialty chemicals, and biofuels (Kuhlman & Bradley, 2019).

In recent years, there has been a surge in the development of protein foundation models[1] aimed at understanding fundamental biological processes by capturing the intricate mechanisms of proteins (Jumper et al., 2021; Abramson et al., 2024; Lin et al., 2023a; Watson et al., 2023b; Ingraham et al., 2023; Krishna et al., 2024; Shin et al., 2021; Madani et al., 2023; Alley et al., 2019; Wang et al., 2024b; Hayes et al., 2024; Hie et al., 2024). These models, leveraging advanced deep-learning and generative AI techniques, have demonstrated remarkable capabilities and mark a significant shift from traditional, task-specific approaches to more generalizable frameworks capable of learning complex patterns and relationships within vast protein datasets. For instance, AlphaFold3 (Abramson et al., 2024), which is based on diffusion models, has achieved unprecedented accuracy in full atom structure prediction for all biomolecules, while others like the ESM series (Rives et al., 2021;

---

[*]Equal contribution.
[†]Corresponding author.
[1]In this study, we broaden the definition of protein foundation models to include any generative model aimed at addressing foundational problems in protein science.

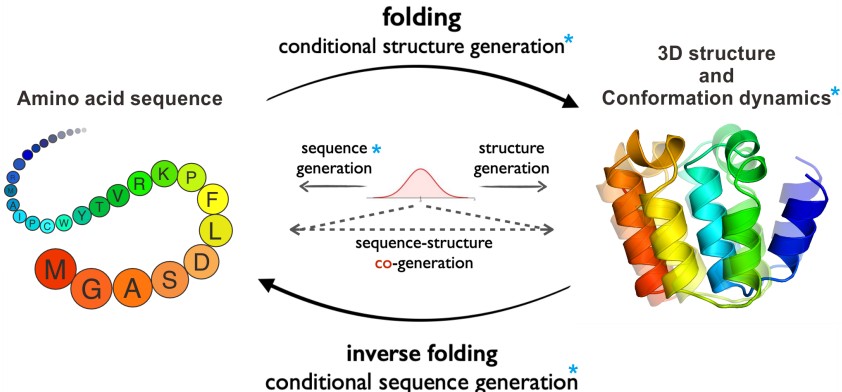

Figure 1: **Comprehensive overview of fundamental protein modeling tasks in ProteinBench.** ProteinBench incorporates a spectrum of protein modeling challenges. Tasks marked with blue stars highlight domains where standardized performance benchmarks were previously unavailable.

Hsu et al., 2022; Lin et al., 2023a; Verkuil et al., 2022; Hayes et al., 2024) and DPLM (Wang et al., 2024b) have shown impressive representation capability in protein language modeling benefiting diverse downstream tasks. Furthermore, these foundation models are not limited to single modalities. Multi-modal models that jointly consider sequence, structure, and function are emerging, offering a comprehensive understanding of protein behavior (Hayes et al., 2024; Liu et al., 2023). In addition to sequence and structural modeling, recent work on protein conformational dynamics further extends model capabilities, offering deeper insights into the connections from sequence to structural dynamics (and ultimately to function) through the lens of generative AI (Jing et al., 2023; Zheng et al., 2024; Jing et al., 2024; Wang et al., 2024c; Lu et al., 2024).

However, the rapid progress of protein foundation models has also led to an urgent need for a unified framework to holistically evaluate their performance across a diverse set of tasks, datasets, and metrics, as shown in Appendix A. The current landscape of protein foundation models is characterized by non-unified modeling approaches, and task-specific or model-specific evaluation criteria. This heterogeneity in evaluation methods makes it challenging to draw meaningful comparisons between different models and to fully understand their relative strengths and limitations.

Through systematic evaluations of datasets spanning diverse biological domains, with a particular emphasis on protein design and conformational dynamics, we aim to provide a comprehensive analysis of model architecture and performance on protein foundation models. This approach allows us to dissect the impact of various model components and data characteristics on different aspects of protein modeling. Comparing the capabilities of these models on standardized benchmarks is crucial for guiding future research directions, informing model selection for practical applications, and driving the advancement of the field as a whole.

In this study, as shown in Figure 1, we present ProteinBench, the first benchmark designed to provide a comprehensive evaluation of protein foundation models through four key components:

**(1) A taxonomic classification of tasks encompassing the main generative challenges in the protein domain.** ProteinBench covers a wide range of generative tasks, including protein design (spanning structure design, sequence design, structure-sequence co-design, and an application-specific task of antibody design) and protein conformation prediction (single-state, multiple-state, and conformational distribution prediction). These tasks, addressing different protein modalities, enable a nuanced analysis of the interplay between model architecture and modal characteristics on performance. We utilize diverse and carefully curated datasets to capture the complexity and diversity of the protein universe, ensuring a thorough evaluation of model capabilities.

**(2) A multi-metric evaluation approach assessing performance across four key dimensions: quality, novelty, diversity, and robustness.** Current evaluations of protein generative models often suffer from non-unified metrics and incomplete assessments, typically focusing on only one or two aspects. However, protein scientific problems encompass a complex and systematic array of challenges. Downstream tasks in protein modeling and design involve intricate interplays between sequence, structure, and function. ProteinBench addresses this limitation by providing a comprehensive measurement of a model's ability based on four critical dimensions: quality, novelty, diversity, and robustness. This multi-faceted approach offers a more holistic view of model performance and capabilities.

Table 1: Overview of ProteinBench, which summarizes the dimensions, metrics, and methods used in ProteinBench. We use '*italics*' for highlighting, a method that has not yet been evaluated in ProteinBench but will be assessed in the future. Details regarding task selection, metric design principles, implementation specifics, and extended discussions are provided in the corresponding sections.

| Tasks | Dimension | Metrics | Methods |
|---|---|---|---|
| ***Protein Design*** | | | |
| Inverse Folding (Section B.1.1) | Sequence recovery Refoldability Stability Robustness | AAR scTM (AF2) pLDDT (AF2) | ProteinMPNN, ESMIF1, LM-Design, ESM3 *PiFold, CarbonDesign* |
| Backbone Design (Section B.1.2) | Quality Novelty Diversity | scTM, scRMSD (ProteinMPNN & ESMFold) Max. TM score to PDB database (Foldseek) Pairwise TM, Max Cluster (Foldseek) | Rfdiffusion, Frameflow, Chroma, Framediff, Foldflow, Genie, Proteus *foldingdiff* |
| Sequence Design (Section B.1.3) | Quality Novelty Diversity | pLDDT (AF2) Max. TM to PDB database (Foldseek) Pairwise TM , Max Cluster (Foldseek) | ProGen2, EvoDiff, DPLM, ESM3 |
| Struct-seq Co-design (Section B.1.4) | Quality Novelty Diversity | scTM, scRMSD (ESMFold) Max. TM score to PDB database (Foldseek) Pairwise TM, Max Cluster (Foldseek) | ProteinGenerator, ProtPardelle, Multiflow, ESM3, CarbonNovo |
| Motif Scaffolding (Section B.1.5) | Quality | Motif RMSD, Scafold RMSD | FrameFlow, Rfdiffusion, TDS, EvoDiff, DPLM, ESM3 |
| Antibody Design (Section B.1.6) | Accuracy Functionality Specificity Rationality | AAR, RMSD, TM-score Binding Energy (Rosetta) Seq Similarity, PHR CN-Score, Clashes, Seq Naturalness Total Energy (Rosetta), scRMSD (IgFold) | HERN, MEAN, dyMEAN, DiffAb, AbDPO |
| ***Protein Conformation Prediction*** | | | |
| Single-state (folding) (Section B.2.1) | Accuracy Quality | TM-score, RMSD, GDT, lDDT CA clash, Peptide bond break | AlphaFold2, OpenFold, ESMFold, RoseTTAFold2, EigenFold |
| Multiple-state (Section B.2.2) | Accuracy Diversity Quality | Ensemble TM-score/RMSD Pairwise RMSD/TM-score CA clash, Peptide bond break | EigenFold, MSA-subsampling, Str2Str, AlphaFlow/ESMFlow, ConfDiff, *Distributed graphormer* |
| Distribution Prediction (Section B.2.3) | Accuracy Diversity Quality | Flexibility accuracy, Distributional similarity, Ensemble observables Pairwise RMSD, RMSF CA clash, Peptide bond break | |

**(3) In-depth analyses from various user objectives, providing a holistic view of model performance.** Recognizing that different users may have varying objectives when applying protein foundation models, we conduct in-depth analyses from multiple perspectives. For instance, in protein design, some users may prioritize models that fit natural evolutionary distributions, while others may seek models capable of generating novel proteins outside the training set distribution. By analyzing model capabilities from these different objectives, ProteinBench provides insights that are beneficial for a wide range of practical applications.

**(4) Leaderboard and code framework.** To facilitate fair comparisons and support the development of new methods, we provide a unified experimental framework. This includes a public leaderboard and open-source code, enabling researchers to easily benchmark their models against existing ones and contribute to the ongoing advancement of the field.

By incorporating these four components, ProteinBench aims to establish a standardized, comprehensive, and user-centric evaluation framework for protein foundation models. This approach not only illuminates the current state-of-the-art but also guides future research directions and accelerates progress in the field of protein modeling and design.

## 2 PROTEINBENCH

In this section, we provide ProteinBench, a holistic evaluation framework for protein foundation models as shown in Table 1, with a particular focus on two key generative tasks: protein design and conformation prediction. These two areas are further divided into eight subtasks. More details about the task definitions, evaluations, and discussions can be found in Appendix B.

### 2.1 PROTEIN DESIGN

We evaluate protein foundation models on six design tasks using standardized metrics, enabling cross-task comparisons and assessment of different modeling approaches.

#### 2.1.1 INVERSE FOLDING

Inverse folding is a fundamental task aiming to design amino acid sequences that can fold into predetermined structures. Evaluations were conducted on different datasets targeting two distinct objectives of structure-based sequence design: evolutionary distribution capturing and de novo protein design. For more detailed information, please refer to Section B.1.1. Evaluation results are presented in Table 2. Our analysis of native distribution fitness reveals that language model-based

Table 2: Performance of structure-based sequence design models on inverse folding tasks. The reported results are the median of repetitive experiments. 'N/A' stands for not applicable. ESMIF1 and ESM3 use all native structures and sequences for model training, therefore, they not measured in the evolution distribution fitting objective.

| Objectives | Fitting Evolution Distribution | | De novo backbones based sequence design | | | | | | | | |
|---|---|---|---|---|---|---|---|---|---|---|---|
| | CASP | CAMEO | length 100 | | length 200 | | length 300 | | length 400 | | length 500 | |
| | AAR ↑ | AAR ↑ | scTM ↑ | pLDDT ↑ | scTM ↑ | pLDDT ↑ | scTM ↑ | pLDDT ↑ | scTM ↑ | pLDDT ↑ | scTM ↑ | pLDDT ↑ |
| ProteinMPNN | 0.450 | 0.468 | **0.962** | **94.14** | **0.945** | **89.34** | **0.962** | **90.28** | **0.875** | **83.76** | **0.568** | **67.09** |
| ESM-IF1 | N/A | N/A | 0.810 | 88.83 | 0.635 | 69.67 | 0.336 | 74.36 | 0.449 | 64.59 | 0.462 | 58.97 |
| LM-DESIGN | **0.516** | **0.570** | 0.834 | 78.45 | 0.373 | 58.41 | 0.481 | 69.86 | 0.565 | 59.87 | 0.397 | 56.35 |
| ESM3 | N/A | N/A | 0.942 | 86.60 | 0.486 | 60.69 | 0.632 | 70.78 | 0.564 | 62.63 | 0.452 | 59.37 |

method LM-DESIGN (Zheng et al., 2023) achieve high sequence recovery rates for native protein structure-based sequence design. This suggests that these models effectively learn and replicate the intricate patterns of amino acid selection that have emerged through evolutionary processes. While its performance decreases when applied to de novo backbone-based sequence design. Conversely, ProteinMPNN (Dauparas et al., 2022a), a method specifically developed for de novo design and trained using coordinates perturbed with 0.2Å added noise, consistently demonstrates superior performance in de novo design tasks. However, ProteinMPNN's performance shows a decline when evaluated on the objective of fitting to native evolution. This finding suggests no single model currently excels across all inverse folding objectives. The choice of model should be carefully aligned with the intended applications.

ESM-IF1 (Hsu et al., 2022), trained on AlphaFoldDB (Varadi et al., 2022) using GVP (Jing et al., 2020) and Transformer architectures with 0.1Å noise, showed suboptimal performance in de novo backbone sequence design. While the model excels at functional mutation prediction as demonstrated in ProteinGYM (Notin et al., 2024), these tasks were not included in our study. The recently released ESM3 (Hayes et al., 2024) performs similarly to ESM-IF1, with advantages at specific sequence lengths (100, 300, and 400 residues). PiFold (Gao et al., 2022) and CarbonDesign (Ren et al., 2024) are not currently included in ProteinBench.

### 2.1.2 STRUCTURE DESIGN

Protein backbone design focuses on generating new protein folds to expand the repertoire of protein structures beyond those found in nature. Further details are provided in Section B.1.2. Evaluation results are presented in Table 3 and Table 12. Based on the quality metrics of scTM-score and scRMSD, RFdiffusion (Watson et al., 2023b) demonstrates exceptional performance in backbone design for chain lengths ranging from 50 to 300 amino acids. FrameFlow (Yim et al., 2023) achieves the second-best performance in this range. However, we observe a significant performance decrease across all models for longer chains (500 amino acids), with scTM scores dropping by more than 20%. This decline suggests that developing methods for long-chain backbone design remains an important challenge for future research. Proteus (Wang et al., 2024a) shows a superior design quality for long-chain backbones (500 residues). However, we observed a significant decline in structure diversity for long chains. Novelty is an equally important metric, as it gauges a method's capacity to explore new structural space beyond known protein folds. Under moderate quality constraints (scTM score >0.5), FoldFlow (Bose et al., 2023) and Genie (Lin & AlQuraishi, 2023) exhibit good performance in generating novel structures. When we increase the quality threshold (scTM score >0.8), Chroma (Ingraham et al., 2023) generally shows the best performance across chain lengths from 50 to 500 amino acids. For structural diversity, Chroma shows commendable performance across the tested chain lengths. It is important to note that we used the released FoldFlow model trained on a smaller training set with shorter sequences. This limitation may lead to an unfair comparison of the model architecture to other methods trained on the entire PDB database, particularly for longer chain lengths. Foldingdiff (Wu et al., 2024a) is not featured in ProteinBench.

### 2.1.3 SEQUENCE DESIGN

Protein sequence design aims to generate amino acid sequences with desired properties. See Section B.1.3 for details. In this section, we assess the performance of various protein sequence generative models. The evaluation metrics include AlphaFold2 (AF2) predicted pLDDT scores for structural plausibility (quality), maximum TM-score and maximum cluster values for structural diversity, and maximum TM-score to PDB structures for structural novelty. We choose representative methods of distinct modeling foundations for evaluation. Among the methods evaluated, ProGen2 (Nijkamp et al., 2023) is an autoregressive protein language model (AR-LM), while EvoDiff (Alamdari et al.,

Table 3: Performance of backbone design models evaluated using various lengths ranging from 50 to 500. The reported results are the median of repetitive experiments. We highlight the **best** performance in bold and the second-best with the underline. For the novelty and diversity metrics, we only highlight results with the corresponding scTM score higher than 0.5. 'N/A' stands for not applicable.

| | length 50 Quality | | length 50 Novelty | length 50 Diversity | | length 100 Quality | | length 100 Novelty | length 100 Diversity | |
|---|---|---|---|---|---|---|---|---|---|---|
| | scTM ↑ | scRMSD ↓ | Max TM ↓ | pairwise TM ↓ | Max Clust. ↑ | scTM ↑ | scRMSD ↓ | Max TM ↓ | pairwise TM ↓ | Max Clust.↑ |
| Native PDBs | 0.91±0.11 | 0.74±1.45 | N/A | 0.29±0.03 | 0.66 | 0.96±0.10 | 0.67±1.61 | N/A | 0.30±0.02 | 0.77 |
| RFdiffusion | **0.95±0.12** | **0.45±1.71** | 0.65±0.16 | 0.58±0.05 | 0.67 | **0.98±0.12** | **0.48±0.56** | 0.76±0.01 | 0.41±0.03 | 0.32 |
| FrameFlow | 0.91±0.09 | 0.58±0.51 | 0.75±0.01 | 0.68±0.10 | 0.39 | 0.94±0.08 | 0.70±0.70 | 0.72±0.01 | 0.55±0.08 | 0.49 |
| Chroma | 0.85±0.15 | 1.05±1.49 | 0.59±0.08 | 0.29±0.01 | 0.48 | 0.89±0.13 | 1.27±1.85 | 0.70±0.01 | 0.35±0.03 | 0.59 |
| FrameDiff(latest) | 0.85±0.13 | 1.00±1.27 | 0.67±0.01 | 0.35±0.02 | 0.64 | 0.90±0.08 | 1.23±1.02 | 0.71±0.08 | 0.52±0.05 | 0.11 |
| FoldFlow1(sfm) | 0.90±0.10 | 0.67±0.88 | 0.68±0.03 | 0.63±0.07 | 0.48 | 0.87±0.11 | 1.34±1.42 | 0.65±0.01 | 0.49±0.08 | 0.83 |
| FoldFlow1(base) | 0.79±0.14 | 1.19±1.27 | 0.66±0.02 | 0.53±0.08 | 0.76 | 0.81±0.15 | 1.70±1.95 | 0.62±0.01 | 0.48±0.07 | 0.83 |
| FoldFlow1(ot) | 0.83±0.16 | 1.10±1.53 | 0.65±0.02 | 0.53±0.08 | 0.77 | 0.83±0.15 | 1.60±1.95 | 0.64±0.01 | 0.48±0.06 | 0.81 |
| Genie | 0.57±0.15 | 3.12±2.07 | **0.57±0.03** | **0.32±0.02** | **0.90** | 0.69±0.17 | 3.38±3.04 | **0.59±0.01** | **0.31±0.02** | **0.96** |
| Proteus | 0.94±0.09 | 0.57±1.15 | 0.78±0.10 | 0.96±0.17 | 0.15 | 0.94±0.12 | 0.84±0.52 | 0.73±0.01 | 0.73±0.08 | 0.5 |

| | length 300 Quality | | length 300 Novelty | length 300 Diversity | | length 500 Quality | | length 500 Novelty | length 500 Diversity | |
|---|---|---|---|---|---|---|---|---|---|---|
| | scTM ↑ | scRMSD ↓ | Max TM ↓ | pairwise TM ↓ | Max Clust. ↑ | scTM ↑ | scRMSD ↓ | Max TM ↓ | pairwise TM ↓ | Max Clust.↑ |
| Native PDBs | 0.97±0.10 | 0.82±2.67 | N/A | 0.28±0.02 | 0.77 | 0.97±0.17 | 1.07±5.96 | N/A | 0.29±0.03 | 0.8 |
| RFdiffusion | **0.96±0.15** | 1.03±3.14 | 0.64±0.01 | **0.36±0.03** | 0.65 | 0.79±0.19 | 5.60±5.66 | 0.62±0.004 | 0.33±0.02 | 0.89 |
| FrameFlow | 0.92±0.15 | 1.95±2.76 | 0.65±0.01 | 0.43±0.07 | 0.88 | 0.61±0.19 | 7.92±4.08 | 0.61±0.01 | 0.40±0.06 | 0.92 |
| Chroma | 0.87±0.13 | 2.47±3.63 | 0.66±0.01 | **0.36±0.04** | 0.67 | 0.72±0.18 | 6.71±5.76 | 0.60±0.01 | **0.29±0.01** | 0.99 |
| FrameDiff(latest) | 0.87±0.12 | 2.73±2.69 | 0.69±0.00 | 0.48±0.04 | 0.21 | 0.63±0.24 | 9.52±18.19 | 0.58±0.03 | 0.40±0.06 | 0.52 |
| FoldFlow1(sfm) | 0.45±0.11 | 9.04±2.52 | 0.54±0.01 | 0.39±0.04 | 1.00 | 0.37±0.06 | 13.04±1.71 | 0.53±0.01 | 0.37±0.03 | 1.00 |
| FoldFlow1(base) | 0.43±0.09 | 9.56±2.42 | 0.54±0.01 | 0.39±0.05 | 0.98 | 0.35±0.05 | 13.20±2.29 | 0.52±0.01 | 0.39±0.05 | 1.00 |
| FoldFlow1(ot) | 0.54±0.12 | 8.21±2.38 | **0.58±0.00** | 0.41±0.06 | **0.94** | 0.37±0.06 | 12.48±2.00 | 0.51±0.01 | 0.35±0.03 | 1.00 |
| Genie | 0.27±0.02 | 20.37±1.70 | 0.30±0.01 | 0.23±0.01 | 1.00 | 0.25±0.01 | 26.08±1.58 | 0.22±0.002 | 0.23±0.004 | 1.00 |
| Proteus | 0.94±0.06 | 1.46±1.08 | 0.78±0.05 | 0.62±0.09 | 0.34 | **0.90±0.13** | **2.76±3.57** | 0.72±0.02 | 0.62±0.09 | 0.34 |

Table 4: Performance of protein sequence generative models/language models on sequence generation tasks. The reported results are the average of repetitive experiments with the standard derivation. The pLDDT score is the output of AlphaFold2. Max TM is an abbreviation for Maximum TM-score to PDB database. 'N/A' stands for not applicable. We highlight the **best** performance in bold.

| | length 100 Quality | | length 100 Diversity | | length 100 Novelty | length 200 Quality | | length 200 Diversity | | length 200 Novelty |
|---|---|---|---|---|---|---|---|---|---|---|
| | ppl ↓ | pLDDT ↑ | pairwise TM ↓ | Max Clust. ↑ | Max TM ↓ | ppl ↓ | pLDDT↑ | pairwise TM ↓ | Max Clust. ↑ | Max TM ↓ |
| Native Seqs | | 68.46±16.50 | 0.55±0.19 | 0.75 | N/A | | 61.91±11.62 | 0.49±0.10 | 0.78 | N/A |
| Progen 2 (700M) | 8.28±3.87 | 64.00±21.26 | **0.42±0.10** | 0.94 | **0.64±0.08** | 5.68±3.64 | 69.91±9.23 | 0.40±0.13 | 0.91 | **0.69±0.05** |
| EvoDiff | 16.89±1.04 | 50.20±10.27 | 0.43±0.05 | **0.98** | 0.69±0.03 | 17.28±1.64 | 50.66±16.38 | 0.36±0.04 | 1.00 | 0.71±0.02 |
| DPLM (650M) | **6.21±3.10** | **85.38±14.20** | 0.50±0.20 | 0.80 | 0.74±0.10 | **4.61±2.63** | **93.54±3.73** | 0.54±0.24 | 0.70 | 0.91±0.004 |
| ESM3 (1.4B) | 14.79±2.90 | 54.26±15.35 | 0.45±0.15 | 0.90 | 0.68±0.07 | 12.96±2.38 | 58.45±9.40 | **0.35±0.07** | **1.00** | 0.80±0.01 |

| | length 300 Quality | | length 300 Diversity | | length 300 Novelty | length 500 Quality | | length 500 Diversity | | length 500 Novelty |
|---|---|---|---|---|---|---|---|---|---|---|
| | ppl ↓ | pLDDT ↑ | pairwise TM ↓ | Max Clust. ↑ | Max TM ↓ | ppl ↓ | pLDDT↑ | pairwise TM ↓ | Max Clust. ↑ | Max TM ↓ |
| Native Seqs | | 61.49±14.47 | 0.51±0.13 | 0.85 | N/A | | 62.95±12.60 | 0.51±0.11 | 0.78 | N/A |
| Progen 2 (700M) | 6.25± 4.02 | 65.69±20.93 | 0.42±0.16 | 0.93 | **0.66±0.06** | 4.27±3.60 | 61.45±20.17 | 0.32±0.11 | 0.95 | 0.68±0.08 |
| EvoDiff | 17.13±2.00 | 45.14±9.95 | **0.31±0.03** | **1.00** | 0.68±0.02 | 16.51±3.82 | 43.14±5.16 | **0.31±0.03** | **1.00** | 0.69±0.02 |
| DPLM (650M) | **3.47±1.44** | **93.07±5.77** | 0.57±0.25 | 0.63 | 0.91±0.01 | **3.33±1.8** | **87.73±11.61** | 0.43±0.18 | 0.85 | 0.85±0.04 |
| ESM3 (1.4B) | 14.59±2.97 | 48.08±13.34 | 0.32±0.03 | **1.00** | 0.75±0.02 | 11.10±2.26 | 52.17±10.52 | 0.30±0.05 | **1.00** | **0.54±0.03** |

2023) is designed as an order-agnostic autoregressive diffusion model (OADM). DPLM (Wang et al., 2024b) and ESM3 (Hayes et al., 2024) share a probabilistic foundation as absorbing discrete diffusion models or generative masked language models. Notably, ESM3 is a multimodal model that advances beyond other sequence-only methods by jointly learning protein sequences, structures and functions through tokenization. For each model and sequence length, we sample 50 sequences to evaluate their performance.

As shown in Table 4, DPLM consistently shows the highest quality scores, indicating superior accuracy in sequence generation. However, it has relatively lower diversity metrics, suggesting less variation in its generated sequences. EvoDiff, while demonstrating lower pLDDT scores, excels in diversity, particularly in producing highly diverse sequence clusters. Surprisingly, ESM3, a multimodal protein LM, displays lower pLDDT in sequence generation, while maintaining competitive diversity, especially in generating novel sequences. ProGen2 strikes a balance between quality and diversity, offering moderate pLDDT scores and satisfactory diversity and novelty. This model is effective for generating sequences that are both diverse and close to known structures, depending on specific application needs. Regarding different chain lengths, all the models generally exhibit consistent trends in their performance metrics. As the chain length increases, there is a slight decline in the quality of sequences generated by some models, particularly for EvoDiff and ESM3.

Table 5: Performance of protein co-design models on structure-sequence co-generation tasks. The reported results are the average of repetitive experiments with the standard derivation. We highlight the best performance in bold and the second-best with the underline. *: We have tried our best to reproduce all models according to the instructions in their respective codebases, using publicly available model weights. However, some results may differ from those reported in the original studies. We welcome any feedback and corrections to help us make timely updates in the future.

| | length 100 | | | | length 200 | | | |
|---|---|---|---|---|---|---|---|---|
| | Quality | | Diversity | Novelty | Quality | | Diversity | Novelty |
| | scTM ↑ | scRMSD ↓ | Max Clust. ↑ | Max TM ↓ | scTM ↑ | scRMSD ↓ | Max Clust. ↑ | Max TM ↓ |
| Native PDBs | 0.91±0.11 | 2.98±3.49 | 0.75 | N/A | 0.88±0.09 | 3.24±3.77 | 0.77 | N/A |
| ProteinGenerator | 0.91±0.08 | 3.75±3.39 | 0.24 | 0.73 | 0.88±0.09 | 6.24±4.10 | 0.25 | 0.72 |
| ProtPardelle | 0.91±0.09 | 2.07±1.87 | 0.73 | **0.16** | 0.92±0.04 | 2.36±1.19 | 0.09 | 0.75 |
| Multiflow | **0.96±0.04** | **1.10±0.71** | 0.33 | 0.71 | **0.95±0.04** | 1.61±1.73 | 0.42 | 0.71 |
| Carbonnovo | 0.91±0.14 | 1.16±1.03 | **0.71** | 0.69 | 0.94±0.09 | **1.18±1.47** | **0.50** | 0.71 |
| ESM3* | 0.72±0.19 | 13.80±10.51 | **0.64** | 0.41 | 0.63±0.20 | 21.18±16.19 | **0.63** | **0.61** |

| | length 300 | | | | length 500 | | | |
|---|---|---|---|---|---|---|---|---|
| | Quality | | Diversity | Novelty | Quality | | Diversity | Novelty |
| | scTM ↑ | scRMSD ↓ | Max Clust. ↑ | Max TM ↓ | scTM ↑ | scRMSD ↓ | Max Clust. ↑ | Max TM ↓ |
| Native PDBs | 0.92±0.12 | 3.94±4.95 | 0.75 | N/A | 0.90±0.14 | 9.64±7.05 | 0.80 | N/A |
| ProteinGenerator | 0.81±0.14 | 9.26±4.13 | 0.22 | **0.71** | 0.41±0.19 | 33.91±15.10 | 0.18 | 0.73 |
| ProtPardelle | 0.94±0.02 | 2.07±0.73 | 0.05 | 0.73 | 0.41±0.10 | 41.10±8.85 | 0.14 | 0.65 |
| Multiflow | **0.96±0.06** | 2.14±3.24 | **0.58** | **0.71** | 0.83±0.15 | 8.48±7.02 | **0.67** | **0.68** |
| Carbonnovo | 0.95±0.08 | **1.33±1.59** | 0.31 | 0.74 | **0.85±0.15** | **4.07±4.14** | **0.67** | **0.68** |
| ESM3* | 0.59±0.21 | 25.5±20.68 | 0.52 | 0.73 | 0.54±0.20 | 33.70±21.08 | 0.37 | 0.77 |

This indicates a challenge in maintaining high sequence quality as the chain length grows. Among them, DPLM demonstrates robust performance across all lengths, maintaining high pLDDT even for longer sequences. Overall, DPLM is good at highly structural protein sequence generation, while EvoDiff and ESM3 are preferable for better diversity and novelty, with ProGen2 offering a balanced performance across metrics.

### 2.1.4 STRUCTURE AND SEQUENCE CO-DESIGN

Protein structure-sequence co-design represents a challenge in protein engineering that involves the simultaneous generation of both backbone structure and amino acid sequence. We evaluate the structural quality, novelty, and diversity similarly to backbone design. See Section B.1.4 for details. in Table 5. We inspect the performance of ProteinGenerator (Lisanza et al., 2023), ProtPardelle (Chu et al., 2024), Multiflow (Campbell et al.), CarbonNovo (Ren et al., 2024) and ESM3 (Hayes et al., 2024) for different lengths. The performance is assessed using metrics similar to those applied in backbone generation. Note that, however, the quality here is about structure-sequence compatibility measuring how well the designed sequence can fold into the corresponding designed structure, using scTM and scRMSD. The key difference is that co-design models are tasked with simultaneously generating both the sequence and structure, while backbone design models require an additional inverse folding model, such as ProteinMPNN, to design the sequence. Other metrics used for evaluation include diversity (max cluster) and novelty (max TM-score to PDB).

As shown in Table 5, ProteinGenerator, ProtPardelle, CarbonNovo, and Multiflow consistently show strong performance of structure-sequence compatibility when length less than 300, with high scTM scores (up to 0.96±0.06) and relatively low scRMSD values, indicating superior structural quality in generated sequences. ProtPardelle and ProteinGenerator particularly excel at shorter lengths while failing at long proteins. CarbonNovo and Multiflow maintain high performance even as sequence length increases, demonstrating their robustness with consistently high scTM scores and lower scRMSD values, which indicates their capability to generate high-quality structures. ESM3, on the other hand, shows suboptimal performance by its low scTM scores and very high scRMSD values, suggesting that it struggles with unconditional generation. Overall, these findings suggest that CarbonNovo and Multiflow are particularly robust and superior as a co-design protein generative model across all tested lengths.

### 2.1.5 MOTIF SCAFFOLDING

Motif scaffolding represents a specialized challenge in protein design that focuses on creating protein structures incorporating specific functional motifs or binding sites. See Section B.1.5 for more details. In this section, we evaluate the performance of various motif-scaffolding methods across dif-

Table 6: Performance of antibody design methods on 55 antibody-antigen complexes from the RAbD dataset. For methods that can generate multiple antibodies (marked with *), the standard deviations between different antibodies generated against the same antigen are also reported.

| | Accuracy | | | Functionality | | Specificity | |
|---|---|---|---|---|---|---|---|
| | AAR ↑ | RMSD ↓ | TM-score ↑ | Binding Energy ↓ | SeqSim-outer ↓ | SeqSim-inner ↑ | PHR ↓ |
| RAbD (natural) | 100.00% | 0.00 | 1.00 | -15.33 | 0.26 | N/A | 45.78% |
| HERN | 33.17% | 9.86 | 0.16 | 1242.77 | 0.41 | N/A | **39.83%** |
| MEAN | 33.47% | **1.82** | 0.25 | 263.90 | 0.65 | N/A | 40.74% |
| dyMEAN | **40.95%** | 2.36* | 0.36 | 889.28 | 0.58 | N/A | 42.04% |
| *dyMEAN-FixFR | 40.05%±1.06 | 2.37±0.03 | 0.35±0.01 | 612.75±56.03 | 0.60 | **0.96** | 43.75%±2.24 |
| *DiffAb | 35.04%±8.36 | 2.53±0.60 | **0.37±0.06** | 489.42±499.76 | 0.45 | 0.40.68%±10.65 | |
| *AbDPO | 31.29%±7.29 | 2.79±3.01 | 0.35±0.06 | **116.06±186.06** | 0.38 | 0.60 | 69.69%±8.49 |
| *AbDPO++ | 36.25%±7.95 | 2.48±0.59 | 0.35±0.06 | 223.73±281.7 | 0.39 | 0.54 | 44.51%±9.55 |
| | Rationality | | | | | | |
| | CN-score ↑ | Clashes-inner ↓ | Clashes-outer ↓ | SeqNat↑ | Total Energy ↓ | scRMSD ↓ | |
| RAbD (natural) | 50.19 | 0.07 | 0.00 | -1.74 | -16.76 | 1.77 | |
| HERN | 0.04 | 0.04 | 3.25 | **-1.47** | 5408.74 | 9.89 | |
| MEAN | 1.33 | 11.65 | 0.29 | -1.83 | 1077.32 | 2.77 | |
| dyMEAN | 1.49 | 9.15 | 0.47 | -1.79 | 1642.65 | **2.11** | |
| *dyMEAN-FixFR | 1.14±1.71 | 8.88±0.55 | 0.48±0.12 | -1.82±0.10 | 1239.29±113.84 | 2.48±0.24 | |
| *DiffAb | 2.02±2.83 | 1.84±1.35 | 0.19±0.31 | -1.88±0.47 | 495.69±350.96 | 2.57±0.77 | |
| *AbDPO | 1.33±2.31 | 4.14±1.84 | 0.10±0.24 | -1.99±0.34 | **270.12±217.45** | 2.79±3.25 | |
| *AbDPO++ | **2.34±3.20** | **1.66±1.28** | **0.08±0.20** | -1.78±0.43 | 338.14±266.48 | 2.50±0.75 | |

ferent scaffolds used in Watson et al. (2023b) and Yim et al. (2024), focusing on their effectiveness in designing scaffold structures. The primary objective of this evaluation is to compare the efficacy of structure-based and sequence-based approaches in generating designable scaffolds. For purely sequence-based methods, e.g., EvoDiff (Alamdari et al., 2023) and DPLM (Wang et al., 2024b), we use ESMFold to predict the structures of their designed motif-scaffold sequences.

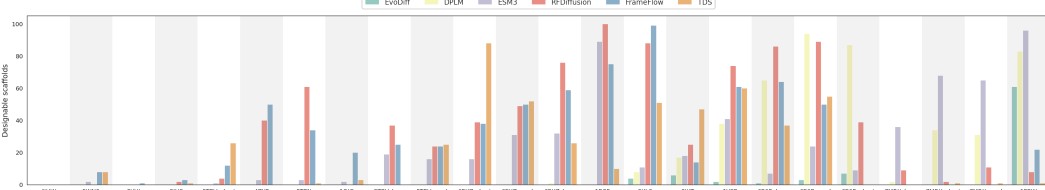

Figure 2: Performance of motif-scaffolding of structure-based and sequence-based methods on the benchmark used in Watson et al. (2023b). Results of FrameFlow, RFDiffusion and TDS are quoted from Yim et al. (2024)[3].

Figure 2 reveals a wide range of performance levels among the tested methods, each exhibiting distinct strengths and weaknesses depending on the specific scaffold context. Notably, structure-based methods such as RFdiffusion (Watson et al., 2023b), TDS (Wu et al., 2024b) and FrameFlow (Yim et al., 2024) consistently perform well across most scenarios, with RFdiffusion showing particular robustness in generating a high number of designable scaffolds. This suggests that structure-based methods are highly effective at capturing the intricate structural details necessary for successful scaffold design. In contrast, sequence-based methods like EvoDiff and DPLM display variable performance, excelling in certain scaffolds that are primarily governed by evolutionary constraints, but underperforming in others with more complex structural motifs. This variability may reflect their limitations in recognizing and adapting to specific structural features.

Interestingly, we find that ESM3 (Hayes et al., 2024), a recent multimodal language model that handles both sequence and structure, performs similarly to advanced structure-based models and even succeeds in cases where those models fail. This shows that multimodal models like ESM3 can process both types of information within a unified framework, making them useful for conditional design. However, ESM3 still doesn't outperform structure-based methods in most cases. Our results highlight that no single model is best for all scaffolds. Future research should focus on combining these methods to leverage their strengths for more effective protein design.

### 2.1.6 ANTIBODY DESIGN

Antibody designing aims to generate antibodies that specifically bind to target antigens (details refer to B.1.6 [Task Definition]). In this section, we selected five antigen-specific antibody design meth-

---

[3]https://github.com/microsoft/protein-frame-flow/tree/main/motif_scaffolding

ods (HERN (Jin et al., 2022), MEAN (Kong et al., 2022), dyMEAN (Kong et al., 2023), DiffAb (Luo et al., 2022), AbDPO (Zhou et al., 2024))and two of their variants (dyMEAN-FixFR and AbDPO++) to evaluate their performance in CDR-H3 generation towards the given antigens. All methods were trained on the same dataset with the configurations reported in the corresponding papers and tested on a unified set of 55 test cases from the RAbD dataset (Adolf-Bryfogle et al., 2018) (further details of model implementation and data construction are provided in Appendix B.1.6 [Model Implementations]) The final results of the evaluation are shown in Table 6. For each evaluation metric, we highlighted the **best** performance in bold and the second-best with the underline, the detailed concept and implementation of each metric can be seen in Appendix B.1.6 [Evaluation Metrics].

In the **Accuracy** evaluation, dyMEAN, and MEAN achieved the best performance in terms of sequence and structure (highest **AAR** and lowest **RMSD**), while DiffAb performed best in **TM-score**.

In the **Functionality** evaluation, all methods produced antibodies with binding energies to the given antigens significantly higher than those of natural antibodies. AbDPO and AbDPO++ achieved the best performance among all the methods.

In the **Specificity** evaluation, we mainly observed the sequence similarity between antibodies against different antigens (**SeqSim-outer**) and the proportion of hydrophobic residues in the generated antibodies (**PHR**). The former metric indicates whether the method can design antibodies specific to a given antigen and DiffAb, AbDPO, and AbDPO++ achieved better performance, while the latter reflects the potential nonspecific binding due to high hydrophobicity and HERN performed best.

In the **Rationality** evaluation, we mainly observed: structural rationality, sequence rationality, and joint structural and sequence rationality. AbDPO++ performed best in structural rationality with fewer clashes and reasonable peptide bond length. HERN performed best in sequence rationality with the highest naturality. AbDPO and dyMEAN perform best in joint structural and sequence rationality from two perspectives. Details of the Specificity and Rationality evaluation refer to Appendix B.1.6 [Extended Explanations and Discussion on Model Performance].

In general, evaluating antibody design methods encompasses various aspects, and using only a few metrics will seriously mislead researchers' understanding of model performance. No single method outperformed all others across the board, and all methods showed substantial gaps compared to natural antibodies. However, AbDPO++, by utilizing synthetic data and aligning with various properties, achieved one of the best performances in almost all metrics among all methods.

## 2.2 PROTEIN CONFORMATION PREDICTION

Protein conformation prediction infers the 3D *conformations* of proteins from their sequences, evaluating models based on their understanding of structure, dynamics, and ultimately functions. We begin by benchmarking common folding models, as they play a crucial role in the development of conformation prediction models. We then compare five recent studies that explore different strategies to extend folding to conformation prediction, focusing on *multiple-state* and *distribution prediction* tasks. A detailed discussion of task definition, evaluation, and results can be found in Section B.2.

### 2.2.1 PROTEIN FOLDING: SINGLE-STATE PREDICTION

Protein folding task predicts the 3D structure of a protein based on its sequence. This sequence-to-structure prediction is a critical measure of a model's understanding of these two modalities. See Section B.2.1 for more details. As shown in Table 7, Multiple Sequence Alignment (MSA)-based folding models (AlphaFold2, OpenFold, RoseTTAFold2) outperform protein language model-based approaches (ESMFold, EigenFold). While the predicted structure quality is comparable among all-atom resolution models, AlphaFold2 and its reproduction, OpenFold, achieve the best performance across all accuracy metrics, offering a strong foundation for conformation prediction.

### 2.2.2 MULTIPLE-STATE PREDICTION

Multiple-state prediction builds upon single-state prediction by aiming to accurately generate two or more distinct conformational states of a protein, typically associated with functional conformational changes. Further details can be found in Section B.2.2. Regarding evaluation, we first investigate the results on **BPTI** with the best model of each study highlighted in Table 8 and the complete evaluation provided in Table 13. The classifier-free guidance in ConfDiff (Wang et al., 2024c) achieved performance comparable to fine-tuning on MD conformation data and, when combined with force guidance, delivered the best overall accuracy (RMSDens). This suggests that incorporating struc-

Table 7: Performance of protein folding on CAMEO2022. Results are reported as mean/median over 183 proteins. The **best** performance is highlighted in bold, and the second-best is underlined. EigenFold only predicts CA coordinates and PepBond break % is not available (shown in "N/A").

| | Accuracy | | | | Quality | |
|---|---|---|---|---|---|---|
| | TM-score ↑ | RMSD ↓ | GDT-TS ↑ | lDDT ↑ | CA clash (%) ↓ | PepBond break (%) ↓ |
| AlphaFold2 | **0.871/0.952** | **3.21**/1.64 | **0.860/0.921** | **0.904/0.933** | **0.3/0.0** | 4.8/4.1 |
| OpenFold | 0.870/0.947 | **3.21/1.59** | 0.856/0.913 | 0.899/**0.933** | 0.4/**0.0** | **2.0/1.7** |
| RoseTTAFold2 | 0.859/0.941 | 3.52/1.75 | 0.845/0.903 | 0.892/0.926 | **0.3/0.0** | 5.5/4.0 |
| ESMFold | 0.847/0.929 | 3.98/2.10 | 0.826/0.881 | 0.870/0.907 | **0.3/0.0** | 4.7/3.4 |
| EigenFold | 0.743/0.823 | 7.65/3.73 | 0.703/0.781 | 0.737/0.810 | 8.0/4.6 | N/A |

Table 8: Performance of multiple-state prediction on BPTI. *Accuracy* metrics (RMSDens, RMSD Cluster 3) are reported as the mean and standard deviation from 20 bootstrap samples at different sample sizes. *Diversity* and *quality* are evaluated based on 1,000 conformations.

| | RMSDens (Å) ↓ | | | RMSD (Å) Cluster 3 ↓ | | | Diversity | Quality | |
|---|---|---|---|---|---|---|---|---|---|
| | N=10 | N=100 | N=1000 | N=10 | N=100 | N=1000 | Pairwise RMSD | CA clash% ↓ | PepBond break%↓ |
| EigenFold | 1.56±0.02 | 1.50±0.01 | 1.46±0.00 | 2.54±0.03 | 2.48±0.01 | 2.46±0.01 | 0.85 | 1.4 | N/A |
| MSA-depth32 | 1.66±0.03 | 1.54±0.04 | 1.41±0.02 | **2.43±0.06** | **2.19±0.16** | **1.85±0.05** | 2.14 | 0.6 | 10.6 |
| Str2Str-ODE ($T_{max} = 0.15$) | 2.40±0.12 | 2.20±0.05 | 2.09±0.01 | 3.00±0.20 | 2.73±0.12 | 2.58±0.05 | 1.86 | **0.0** | 13.9 |
| ESMFlow-MD | 1.68±0.06 | 1.47±0.04 | 1.39±0.03 | 2.44±0.11 | 2.27±0.10 | 2.18±0.02 | 1.17 | **0.0** | 14.3 |
| ConfDiff-ESM-Force | **1.58±0.04** | **1.43±0.03** | **1.36±0.01** | 2.44±0.06 | 2.35±0.05 | 2.24±0.06 | 1.76 | 0.1 | **8.9** |

tural exploration and physical constraints can enhance the sampling of high-accuracy conformations. However, structural exploration alone may be error-prone as Str2Str (Lu et al., 2024), the structure-only models, showed low accuracy even with small perturbations. Other strategies to enhance conformation sampling also showed improved performance: MSA subsampling (Del Alamo et al., 2022) with reduced MSA depth excelled at sampling Cluster 3, the most difficult state to capture, and ESMFlow (Jing et al., 2024) fine-tuned on the MD dataset showed improved diversity and accuracy compared to the PDB-trained base model. However, these approaches also experienced a decline in quality, with increased CA clashing or peptide bond breaking. Lastly, for most methods, increasing the sample depth ($N$) significantly improved expected accuracy, suggesting that a sufficient sample size is essential for thorough evaluation.

***Apo-holo*** is a more challenging dataset where models are required to predict both the unbound (*apo*) and bound (*holo*) conformations induced by ligand binding. As shown in Table 9 and Table 14, strategies to enhance conformation diversity did not improve the TMens score, and the best-performing models closely resemble folding models. In comparison, a baseline model that consistently predicts the perfect *apo* structure achieved a higher TMens score. These findings suggest these models struggle to accurately sample *apo-holo* conformation changes, and higher accuracy may stem from using a stronger folding model.

In summary, strategies such as MSA subsampling, guidance during diffusion, or training on MD conformation data can improve sample diversity and enhance ensemble accuracy for certain local dynamics (as in BPTI). However, they may not be sufficient to capture the complex dynamics involved in processes like *apo-holo* conformational changes.

### 2.2.3 DISTRIBUTION PREDICTION

Distribution prediction assesses a model's ability to generate distributions that closely resemble a target distribution, such as the empirical distribution obtained from molecular dynamics (MD) simulations. See Section B.2.3 for details. The results are summarized in Table 10 and Table 15. We include reference performances for (1) MD iid: independent samples from reference MD trajectories and (2) MD 2.5 ns: consecutive samples from the trajectories corresponding to 2.5 ns of simulation.

Overall, generative models trained to sample protein conformations (AlphaFlow, ConfDiff) significantly outperform perturbation-based methods (MSA subsampling and Str2Str), regardless of perturbation levels. Using a strong folding model like AlphaFold2 generally results in higher accuracy. Classifier-free guidance in ConfDiff improved distribution sampling but was less effective than direct fine-tuning on MD data, highlighting the importance of large-scale conformational data for future models. Additionally, the trade-offs between diversity, prediction performance, and sample quality persist: fine-tuning on MD data improves sample diversity and performance for AlphaFlow but decreases sample quality.

AlphaFlow and ConfDiff models fine-tuned on MD data have shown promise in capturing the conformational distributions, achieving performance comparable to that of 2.5 ns MD simulations on

Table 9: Performance of multiple-state prediction on *apo-holo*. *apo/holo*-TM represents the maximum TM-score of the samples relative to the reference *apo/holo* structure. 20 conformations were sampled for each protein, and the results are reported as mean across 91 proteins.

| | Accuracy | | | Diversity | Quality | |
|---|---|---|---|---|---|---|
| | *apo*-TM ↑ | *holo*-TM ↑ | TMens ↑ | Pairwise TM | CA clash % ↓ | PepBond break % ↓ |
| *apo* model | 1.000 | 0.790 | 0.895 | N/A | N/A | N/A |
| EigenFold | 0.831 | 0.864 | 0.847 | 0.907 | 3.6 | N/A |
| MSA-depth256 | 0.845 | 0.889 | 0.867 | 0.978 | **0.2** | **4.6** |
| Str2Str-ODE ($T_{max} = 0.3$) | 0.766 | 0.781 | 0.774 | 0.872 | **0.2** | 14.7 |
| AlphaFlow-PDB | **0.855** | **0.891** | **0.873** | 0.924 | 0.3 | 6.6 |
| ConfDiff-Open-PDB | 0.847 | 0.886 | 0.867 | 0.909 | 0.5 | 5.5 |

Table 10: Performance on distribution prediction for ATLAS. 250 conformations were sampled for each protein and the median values across 82 proteins are reported. *These metrics are not available for models that lack side-chain or full backbone information.*

| | Diversity | | Flexibility: *Pearson r* on | | | Distributional accuracy | | | |
|---|---|---|---|---|---|---|---|---|---|
| | Pairwise RMSD | *RMSF | Pairwise RMSD ↑ | *Global RMSF ↑ | *Per target RMSF ↑ | *RMWD ↓ | MD PCA W2 ↓ | Joint PCA W2 ↓ | PC sim > 0.5 %↑ |
| MD iid | 2.76 | 1.63 | 0.96 | 0.97 | 0.99 | 0.67 | 0.73 | 0.71 | 93.9 |
| MD 2.5ns | 1.54 | 0.98 | 0.89 | 0.85 | 0.85 | 2.22 | 1.55 | 1.89 | 36.6 |
| EigenFold | 5.96 | N/A | -0.03 | N/A | N/A | N/A | 2.31 | 7.96 | 12.2 |
| MSA-depth256 | 0.83 | 0.53 | 0.25 | 0.34 | 0.59 | 3.60 | 1.79 | 2.91 | 29.3 |
| Str2Str-ODE ($T_{max} = 0.14$) | 1.66 | N/A | 0.13 | N/A | N/A | N/A | 2.14 | 4.39 | 6.1 |
| AlphaFlow-MD | 2.87 | 1.63 | 0.53 | 0.66 | **0.85** | **2.64** | 1.55 | 2.29 | **39.0** |
| ConfDiff-Open-MD | 3.43 | 2.21 | **0.59** | **0.67** | **0.85** | 2.75 | **1.41** | **2.27** | 35.4 |

| | Ensemble observables | | | | Quality | | |
|---|---|---|---|---|---|---|---|
| | Weak contacts $J$ ↑ | Transient contacts $J$↑ | *Exposed residue $J$ ↑ | *Exposed MI matrix $\rho$ ↑ | CA clash % ↓ | *PepBond break % ↓ | |
| MD iid | 0.90 | 0.80 | 0.93 | 0.56 | 0.0 | 3.4 | |
| MD 2.5ns | 0.62 | 0.45 | 0.64 | 0.25 | 0.0 | 3.4 | |
| EigenFold | 0.36 | 0.19 | N/A | N/A | 5.6 | N/A | |
| MSA-depth256 | 0.30 | 0.29 | 0.36 | 0.06 | **0.0** | **5.5** | |
| Str2Str-ODE ($T_{max} = 0.14$) | 0.42 | 0.18 | N/A | N/A | **0.0** | 12.1 | |
| AlphaFlow-MD | 0.62 | **0.41** | **0.69** | **0.35** | **0.0** | 22.2 | |
| ConfDiff-Open-MD | **0.63** | 0.39 | 0.65 | 0.33 | 0.5 | 6.5 | |

some metrics. However, a key goal for these models is to achieve i.i.d. sampling equivalent to long-term MD simulations, and benchmark results reveal a remaining gap in reaching this objective.

## 3 CONCLUSIONS AND FUTURE WORK

In summary, we present the first comprehensive study evaluating the capabilities of various protein foundation models across eight distinct tasks, with a particular focus on protein design and conformation dynamics. We have developed a unified, multi-metric evaluation framework, which is essential for unbiased assessment of protein foundation models from multiple facets. Based on the performance results, we provide insights and considerations for the development and effective use of protein foundation models, offering guidance for future research.

With the detailed discussion available in the Appendix C, we highlight the key observations from our holistic evaluation as follows: (**1**) Valid evaluation of protein foundation models requires accurate and comprehensive evaluation metrics; (**2**) No single model currently excels across all protein design objectives. The choice of model should be carefully aligned with the intended applications; (**3**) While generative models extended from classic folding models have shown the ability to sample protein conformations, challenges remain in both multiple-state prediction and distribution prediction.

### LIMITATIONS AND FUTURE WORK

We acknowledge several limitations and opportunities for enhancement in our current benchmark: (1) The selection of foundation models may not be exhaustive. Future iterations should incorporate additional foundation models to provide a more comprehensive comparison. (2) Inconsistencies in training data across models currently hinder direct comparisons of different model architectures. Future work could address this by standardizing datasets, allowing for more accurate comparisons of architectural performance. (3) The benchmark could be expanded to include a wider range of tasks, further broadening its scope and utility. We are committed to continually refining and expanding ProteinBench. Our vision is for it to evolve into a dynamic, growing benchmark that accelerates progress in the field of protein modeling and design.

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

# A OVERVIEW OF PROTEIN FOUNDATION MODEL BENCHMARKS

## A.1 GENERAL BENCHMARK DESIGN RATIONALE

The field of protein three-dimensional structure prediction has witnessed remarkable progress, exemplified by established benchmarks like CASP and CAMEO, and breakthrough methodologies including AlphaFold series, RosettaFold, ESMFold, and OmegaFold. While structure prediction focuses on determining protein structures from known sequences, protein design addresses the inverse challenge: creating sequences that will fold into desired structures or achieve specific functions. Despite growing interest in protein design, the field has been hampered by the absence of a comprehensive benchmark, with existing evaluations primarily targeting specialized tasks, as documented in Appendix Table 1. A similar limitation exists in conformational dynamics research. Our work addresses these gaps by introducing the first comprehensive benchmark focusing on protein design and conformation prediction.

In our benchmark, protein design is categorized into five distinct areas, following the natural sequence-structure-function hierarchy. This begins with sequence design, focusing on optimizing amino acid sequences for stable folding, and progresses to backbone design, which involves engineering the overall protein architecture. The more complex sequence-structure co-design task requires simultaneous optimization of both sequence and structure. At the functional level, motif scaffolding involves incorporating functional motifs into stable scaffolds, while antibody design represents a specialized application focusing on engineering antibody structures and sequences for antigen binding, particularly crucial for therapeutic development.

The conformation prediction component is similarly structured into three distinct categories, reflecting increasing levels of complexity in protein dynamics. Single conformation prediction focuses on identifying the lowest energy state among possible conformations. Multiple conformation prediction addresses the more complicated challenge of predicting discrete conformational states. The most sophisticated category, conformational distribution prediction, tackles the complex task of predicting probability distributions of conformations, essential for understanding proteins with dynamic structural ensembles.

## A.2 OVERVIEW OF EXISTING BENCHMARKS

In this section, we provide a comprehensive overview of existing benchmarks for protein foundation models. Table 11 illustrates the current landscape of these benchmarks, revealing significant limitations in the scope and applicability. The majority of existing benchmarks are narrowly focused, primarily addressing task-specific evaluations rather than offering a holistic assessment of protein foundation models.

Notably, our proposed ProteinBench stands out by offering the most comprehensive coverage across various tasks. It encompasses a wide range of evaluations, including inverse folding, backbone

design, sequence design, structure-sequence co-design, and antibody design in the protein design domain, as well as single-state folding, and multiple-state prediction in the conformational dynamics domain.

Table 11: A comparison of benchmarks for protein fundamental tasks.

| Benchmark | Protein Design | | | | | | Protein Conformation Prediction | | |
|---|---|---|---|---|---|---|---|---|---|
| | Inverse Folding | Backbone Design | Sequence Design | Struc-Seq Codesign | Motif scaffolding | Antibody Design | Folding (single-state) | Multiple State Prediction | Distribution Prediction |
| PDB-Struct (Wang et al., 2023) | ✓ | ✗ | ✗ | ✗ | ✗ | ✗ | ✗ | ✗ | ✗ |
| Proteininvbench (Gao et al., 2024) | ✓ | ✗ | ✗ | ✗ | ✗ | ✗ | ✗ | ✗ | ✗ |
| RFDiffusion (Watson et al., 2023b) | ✗ | ✗ | ✗ | ✗ | ✓ | ✗ | ✗ | ✗ | ✗ |
| CASP (cas, 2022) | ✗ | ✗ | ✗ | ✗ | ✗ | ✗ | ✓ | ✗ | ✗ |
| CAMEO (Robin et al., 2021) | ✗ | ✗ | ✗ | ✗ | ✗ | ✗ | ✓ | ✗ | ✗ |
| PINDER (Kovtun et al., 2024) | ✗ | ✗ | ✗ | ✗ | ✗ | ✗ | ✓ | ✗ | ✗ |
| ProteinBench | ✓ | ✓ | ✓ | ✓ | ✓ | ✓ | ✓ | ✓ | ✓ |

# B    DETAILS ON BENCHMARKING EVALUATIONS

In this section, we provide detailed discussions for each task addressed by various protein foundation models, as shown in Table 1. Our focus will be on the following aspects:

[Task Definition] A detailed description of the task, including its objectives and relevance to protein science. Specification of the input data format and expected output for each task. The impact of the task for protein is provided.

[Evaluation Metrics] Justification and description of the metrics used to assess model performance, including quality, novelty, diversity, and robustness measures. For each specific task, we provided a detailed thought process behind the metric selection and detailed implementation information.

[Datasets] Overview of the datasets used for each task, including their size, diversity, and any pre-processing steps applied. Detailed considerations of datasets, such as dataset impacts, are provided.

[Model Implementations] We place the detailed implementation information for the evaluated methods in this part.

[Extended Explanations and Discussion on Model Performance] Due to space limitations in the main text, we provide an additional explanation and discussion of the performance of specific methods here.

## B.1    PROTEIN DESIGN

### B.1.1    INVERSE FOLDING

[Task Definition] Inverse folding is a fundamental task in protein engineering aimed at designing amino acid sequences that can fold into predetermined structural configurations. This task is essential for various applications, including capturing evolutionary distribution of protein sequences (Zheng et al., 2023), facilitating de novo protein design (Dauparas et al., 2022b), and optimizing protein stability for therapeutic and industrial purposes (Sumida et al., 2024).

[Evaluation Metrics] A key factor in inverse folding is the diverse range of downstream objectives, each requiring specific datasets and evaluation approaches. To provide a comprehensive assessment, we evaluate model performance separately for these distinct objectives. For assessing evolutionary distribution capture, we employ datasets comprising native protein structures. To evaluate sequence design capabilities for novel structures, we utilize datasets of protein backbones generated by RFdiffusion (Watson et al., 2023). Additionally, we incorporate the pLDDT metric to evaluate the predicted folding stability of designed sequences, providing insights into their structural reliability.

For the inverse folding task, metrics for evaluation have evolved over time. ProteinMPNN initially employed amino acid recovery (AAR) as the primary evaluation metric(Dauparas et al., 2022a). More recent studies have introduced structure-based self-consistency TM-score (scTM) as an additional metric for this task  (Gao et al., 2024; Ren et al., 2024). Based on these previous studies, we recognize that both metrics provide valuable and complementary insights into model performance. AAR measures sequence-level accuracy while scTM assesses the structural validity of the designed sequences. Performance in protein sequence design is assessed using multiple complementary metrics.

- **Sequence Recovery** Amino Acid Recovery Rate (AAR), measures the sequence recovery rate of designed proteins and quantifies how well the design method can recapitulate evolutionarily conserved sequence patterns associated with specific structural motifs. While it is highly efficient for evolutionary design evaluation, AAR is limited to cases with known ground truth sequences and cannot assess the refoldability of de novo designed sequences.

- **Refoldability** self-consistency TM-score (scTM), assesses the structural refoldability of designed sequences and has been widely adopted in the field for structural validation.This measure evaluates the structural similarity between the target backbone and the predicted structure of the designed sequence. scTM can be applied to both native and de novo designed sequences, making it more versatile. The prediction is performed using AlphaFold2 (Jumper et al., 2021). We used the Co-labFold implementation Mirdita et al. (2022) for AlphaFold2 inference, with input MSAs obtained using the Colab pipeline. Specifically, the designed sequences are input into AlphaFold2 in the MSA alignment mode. For each prediction round, five structures are generated, and the candidate structure with the highest pLDDT confidence score is selected for comparison with the targeted structure. Similarity is quantified using self-consistency template modeling score (scTM) (Trippe et al., 2022) and self-consistency root-mean-square deviation (scRMSD), providing insight into how well the designed sequence would fold into the intended structure.

- **Stability** Stability is assessed using the predicted local distance difference test (pLDDT), which is also calculated by AlphaFold2 in the same MSA alignment mode used for the scTM score. The pLDDT score serves as a proxy for the predicted stability of the designed protein, as utilized in Dauparas et al. (2022a). We have observed that removing the MSA alignment may yield optimal results for pLDDT, and we plan to update our findings accordingly in future analyses.

**[Datasets]** Typically, inverse folding methods are trained using CATH or PDB datasets. To prevent data leakage, we utilized newly released PDB structures collected from CASP and CAMEO, as well as de novo designed backbones that have not been included in any training sets. Evaluations were conducted on different datasets targeting two distinct objectives of structure-based sequence design.

- **Capture the native evolutionary distribution** We evaluated two independent datasets containing newly released experimentally well-determined PDB structures: CASP15 (cas, 2022) and CAMEO (Robin et al., 2021). We collected new structures from the ongoing CAMEO assessment between January and July 2024, resulting in a total of 332 complex structures. Additionally, 32 protein structures were collected from CASP15, which includes only protein entities, excluding nucleic acids or ligands. These datasets mainly comprise native single-chain protein structures and sequences, making it optimal for evaluating evolutionary information capture and enabling assessment of natural sequence recovery.

- **De novo protein design** RFdiffusion (Watson et al., 2023a) was used to unconditionally generate backbones of varying lengths: specifically, 100, 200, 300, 400, and 500 residues. For each length, 10 different structures were randomly sampled, using a sampling temperature of 0.1 for all methods. The designability of these sequences was evaluated using AlphaFold2, with the scTM score and pLDDT metrics serving as the primary assessment criteria. Existing benchmarks for inverse folding, such as PDB-Struct (Wang et al., 2023) and Proteininvbench (Gao et al., 2024), provide standardized protein structure sets for evaluating inverse folding methods. While these benchmarks have significantly contributed to the field's advancement, there is a growing need for more comprehensive evaluation frameworks. These expanded evaluations should align more closely with diverse user objectives in protein design, encompassing aspects like accuracy in capturing natural evolutionary distributions and robustness in de novo backbone-based sequence design.

**Model implementations** A sampling temperature of 0.1 was used for each method to generate sequences. While this value balances sequence diversity and quality, optimal temperatures may vary across inverse folding methods. For each structure, one generated sequence was predicted for performance evaluation.

- ProteinMPNN (Dauparas et al., 2022a): We follow the official repository and instructions for inference, with the sampling temperature set to 0.1. The default model weight `v_48_020.pt` is used.

- ESM-IF1 (Hsu et al., 2022): We use the public ESM repository for inference.

- LM-DESIGN (Zheng et al., 2023): We used the official repository with the model `lm_design_esm2_650m`.

- ESM3 (Hayes et al., 2024): We follow the official repository with model `esm3-open-small`, which has 1.4B parameters.

**[Extended Explanations and Discussion on Model Performance]**

- **Dataset Characteristics Impact Performance** Our evaluation spans two dataset types: high-quality experimental structures (CASP/CAMEO) and computational de novo structures containing inherent noise. Models performing well on the more challenging de novo structures demonstrate superior robustness, as they must overcome structural uncertainties while maintaining design accuracy.

- **Training Strategy Influences Robustness** ProteinMPNN's approach of incorporating backbone noise during training proves highly effective. Our results confirm their findings that increased training noise correlates with improved model robustness. This is evidenced by ProteinMPNN's superior performance in de novo backbone-based sequence design, validating backbone noise augmentation as an effective strategy for enhancing model resilience.

### B.1.2 PROTEIN BACKBONE DESIGN

**[Task Definition]** Protein backbone design is a classical protein design problem, centered on developing new protein folds to meet de novo design objectives. It is widely accepted that novel structures can approximate new functions. As such, the backbone design task is crucial for expanding the repertoire of protein structures beyond those found in nature, offering significant applications in areas such as drug discovery, biomaterials, and therapeutics. With de novo designed protein backbones, inverse folding methods can be employed to generate the corresponding sequences.

**[Evaluation Metrics]** The rapid development of protein backbone design methods in recent years has been accompanied by inconsistent evaluation practices across different studies. While some researchers report quality in terms of designability, others utilize scTM scores, making it difficult to draw meaningful comparisons between methods. To address this fragmentation in the field, our benchmark implements a comprehensive evaluation framework with standardized metrics. This unified approach enables both fair comparisons between methods and thorough assessment of their performance across multiple aspects of backbone design quality.

Building on previous studies (Trippe et al., 2022; Lin & AlQuraishi, 2023; Yim et al., 2023; Watson et al., 2023b), we evaluate backbone design through multiple criteria that assess the quality, diversity, and novelty of generated structures.

- **Quality** We utilize ProteinMPNN (Dauparas et al., 2022a) to generate eight different sequences for each backbone structure, which are unconditionally sampled from various methods. The structures of these eight sequences are predicted using ESMFold (Lin et al., 2023a). We employ self-consistency TM-score (scTM) and self-consistency RMSD (scRMSD) to measure structural refoldability. For each backbone, we select the sequence with the maximum scTM score. For each method, we unconditionally sample 100 different backbones and report the median scTM and scRMSD.

- **Novelty** Equally important are novelty metrics, which gauge the method's capacity to explore new structural space beyond known protein folds. The metrics measuring the novelty of generated structures were introduced in recent studies (Yim et al., 2023; Campbell et al.). This aspect is evaluated using two key metrics: The maximum TM-score obtained when comparing designed structures to existing entries in the RCSB Protein Data Bank (PDB) (Berman et al., 2000). This comparison is performed using Foldseek with a threshold TM-score of less than 0.5 (van Kempen et al., 2022).

- **Diversity** Two metrics were used to evaluate diversity: (a) We calculate the average pairwise maximum TM-scores among the designed structures. A lower TM-score indicates better diversity. (b) The number of distinct structural clusters identified within the set of designed backbones was also determined using Foldseek (van Kempen et al., 2022) with a TM-score threshold less than 0.5. More clusters stand for higher diversity. These diversity metrics help quantify the range of

Table 12: Performance of backbone design models evaluated using 200, and 400 amino acids in lengths. The reported results are the medium of repetitive experiments. We use bold text to highlight the best and suboptimal results for each metric. For the novelty and diversity metrics, we only highlight results with the corresponding scTM score higher than 0.5.

| | length 200 | | | | | length 400 | | | | |
| | Quality | | Novelty | Diversity | | Quality | | Novelty | Diversity | |
| | scTM ↑ | scRMSD ↓ | Max TM ↓ | pairwise TM ↓ | Max Clust. ↑ | scTM ↑ | scRMSD ↓ | Max TM ↓ | pairwise TM ↓ | Max Clust.↑ |
|---|---|---|---|---|---|---|---|---|---|---|
| Native PDBs | 0.974 | 0.674 | N/A | 0.278 | 0.790 | 0.970 | 1.085 | N/A | 0.261 | 0.840 |
| RFdiffusion | **0.982** | **0.617** | 0.638 | 0.363 | 0.64 | **0.927** | **2.12** | 0.634 | **0.356** | 0.720 |
| FrameFlow | **0.953** | **1.02** | 0.648 | 0.458 | 0.800 | **0.805** | **4.46** | 0.620 | 0.4196 | **0.920** |
| Chroma | 0.892 | 1.776 | 0.674 | 0.346 | 0.620 | 0.761 | 4.891 | **0.626** | **0.304** | **0.95** |
| FrameDiff (latest) | 0.893 | 1.789 | 0.689 | 0.464 | 0.260 | 0.800 | 4.324 | 0.668 | 0.467 | 0.330 |
| FoldFlow1 (base) | 0.529 | 7.108 | 0.579 | 0.430 | **0.950** | 0.415 | 11.743 | 0.525 | 0.357 | 1.00 |
| FoldFlow1 (sfm) | 0.619 | 5.270 | **0.586** | 0.433 | **0.980** | 0.398 | 11.135 | 0.534 | 0.372 | 0.99 |
| FoldFlow1 (ot) | 0.528 | 6.877 | **0.582** | **0.392** | 0.900 | 0.418 | 10.78 | 0.559 | 0.365 | 0.99 |
| Genie | 0.367 | 13.699 | 0.431 | 0.264 | 1.00 | 0.251 | 24.453 | 0.238 | 0.229 | 1.00 |

unique structures the design method can produce, ensuring that it's not simply recreating known folds but generating a varied repertoire of protein backbones.

**[Datasets]** Protein backbone design is a generative task, whose primary objective is to map the overall distribution of the training set accurately. To conduct a comprehensive performance analysis of protein structure generation, we compare the models' performance against the distribution of real PDB structure data. Specifically, we randomly sampled 100 high-resolution experimental structures from the Protein Data Bank (PDB) to serve as references for the data distribution. To ensure diversity, we iteratively removed structures with the highest TM-score compared to others until we arrived at a final set of 100 distinct structures. This approach provides a representative snapshot of the single-chain structural distribution within the PDB, serving as a benchmark for evaluating the performance of generative models in capturing the true distribution of protein structures.

**[Model implementations]** To ensure comprehensive evaluation, we test each method's ability to perform unconditional monomer generation across diverse protein sizes. We follow the official repository and instructions for inference. The generation process is constrained only by specified target lengths, which we set at 50, 100, 200, 300, 400, and 500 residues. This range allows us to assess method performance across both small proteins and larger, more complex structures.

**[Extended Explanations and Discussion on Model Performance]** A notable observation across different backbone design methods reveals an inverse relationship between structural quality and diversity: as methods generate more diverse and novel structures, the quality of the generated backbones tends to decrease. We emphasize that structural quality should be considered the primary metric, as diversity and novelty become meaningful only when the generated structures maintain sufficient quality. Without adequate structural quality, high diversity or novelty scores may simply reflect the generation of unrealistic or physically implausible conformations.

In the revised manuscript, we have expanded our evaluation to include Proteus's performance across multiple protein lengths (100, 200, 300, and 500 residues). Our analysis reveals that Proteus demonstrates superior design quality for long-chain backbone design (500 residues), achieving a scTM score of 0.90 compared to RFdiffusion's 0.79. However, we observed a significant decline in structural diversity for Proteus when designing longer chains. At 300 residues, Proteus shows a diversity score of 0.34 vs. RFdiffusion of 0.65. At 500 residues, Proteus shows a diversity score of 0.34 vs. RFdiffusion of 0.89. Case analysis revealed that Proteus tends to generate structures limited to three categories, predominantly characterized by helical tandem repeats, confirming our diversity metric findings.

**[Additional results for backbone design]**

In this section, we provide more detailed evaluation results of protein backbone design across additional lengths 200 and 400. The results are shown in Table.12.

### B.1.3 PROTEIN SEQUENCE DESIGN

**[Task Definition]** Protein sequence design aims to generate amino acid sequences with desired properties, including quality, diversity, and novelty. The task encompasses both sequence-level

evaluation and assessment of structural characteristics of the generated sequences, making it fundamental for various applications in protein engineering and therapeutic development.

**[Evaluation Metrics]** The evaluation of protein sequence design requires a multi-faceted approach that considers both sequence-level properties and structural characteristics. Sequence-level assessment should verify that generated sequences follow natural protein patterns, while structural evaluation ensures the designs are likely to fold into stable, well-defined conformations. This comprehensive evaluation strategy helps validate both the theoretical and practical utility of designed sequences.

The evaluation of protein sequence design requires a multi-faceted approach that considers both sequence-level properties and structural characteristics. Sequence-level assessment should verify that generated sequences follow natural protein patterns, while structural evaluation ensures the designs are likely to fold into stable, well-defined conformations. This comprehensive evaluation strategy helps validate both the theoretical and practical utility of designed sequences.

We employ multiple complementary metrics to assess different aspects of sequence design:

- **Sequence Naturalness** We utilize perplexity scores from ProGen2 (Nijkamp et al., 2023), an autoregressive protein language model, to quantify how well the generated sequences align with natural sequence distributions. Lower perplexity indicates sequences that better match patterns observed in natural proteins.

- **Structural Stability** For structure-based evaluation, we use single-sequence folding model ESM-Fold (Lin et al., 2023b) to predict the structure of the generated sequences, and then measure the structural quality by pLDDT as the proxy of structural stability of the sequence.

- **Structural Properties** We evaluate structural diversity and novelty using the same protocols established in backbone design evaluation, ensuring a comprehensive assessment of the structural characteristics of generated sequences.

**[Datasets]** For the training dataset, UniRef50 (Suzek et al., 2015) is the commonly used dataset for training protein sequence generative models and language models. Similar to backbone design, to conduct a comprehensive performance analysis of protein sequence generation, we compare the models' performance against the distribution of real sequence data. Specifically, we randomly sampled 50 protein sequences from UniRef50 to serve as references for the data distribution.

**[Model implementations]**

To ensure comprehensive evaluation, we test each method's ability to perform unconditional monomer generation across diverse protein sizes. We follow the official repository and instructions for inference. The generation process is constrained only by specified target lengths, which we set at 100, 200, 300, 400, and 500 residues.

### B.1.4 STRUCTURE AND SEQUENCE CO-DESIGN

**[Task Definition]** Protein structure-sequence co-design represents an advanced challenge in protein engineering that involves the simultaneous optimization of both backbone structure and amino acid sequence. This integrated approach aims to achieve desired properties or functions while maintaining structural stability and sequence compatibility. Unlike isolated sequence or structure design, co-design explores a significantly larger solution space, making it both more challenging and potentially more powerful for creating novel functional proteins.

**[Evaluation Metrics]** The evaluation of co-designed proteins requires a comprehensive framework that addresses both structural and sequence aspects simultaneously. This dual consideration is essential because success in either domain alone is insufficient - the designed sequence must be compatible with its intended structure, and both must contribute to the desired functional properties. The interdependence of sequence and structure necessitates metrics that can capture this relationship effectively.

We implement a multi-faceted evaluation approach that combines metrics from both sequence and structure design domains:

- **Quality** We use a similar self-consistency-based designability in backbone design for the co-design task, but with a key difference. The quality for simultaneously co-generated structure and sequence is about structure-sequence compatibility by measuring how well the designed sequence can fold into the corresponding designed structure, using scTM and scRMSD, whereas backbone design models require an additional inverse folding model, such as ProteinMPNN, to design the sequence.

- **Novelty** Similar to backbone design, we measure the structural novelty compared to known proteins, ensuring that designs represent meaningful additions to the protein design space while maintaining realistic properties.

- **Diversity** Similar to backbone design, we measure the structural diversity by maximum pair-wise TM-scores and the number of distinct structural clusters using Foldseek.

[**Datasets**] High-resolution protein structures from the Protein Data Bank (PDB) are the commonly used datasets for this task, with careful consideration given to remove redundancy.

[**Model implementations**] **To ensure comprehensive evaluation, we test each method's ability to perform unconditional monomer generation across diverse protein sizes. We follow the official repository and instructions for inference. The generation process is constrained only by specified target lengths, which we set at 100, 200, 300, 400, and 500 residues. This range allows us to assess method performance across both small proteins and larger, more complex structures.**

### B.1.5 MOTIF SCAFFOLDING

[**Task Definition**] Motif scaffolding represents a specialized challenge in protein design that focuses on creating protein structures incorporating specific functional motifs or binding sites. The objective is to engineer a stable protein framework that precisely positions the desired motif while maintaining its functional geometry. This task is crucial for developing proteins with targeted functionalities, including enzyme design, therapeutic proteins, and biomolecular recognition systems.

[**Evaluation Metrics**] Evaluating motif scaffolding designs requires a careful balance between maintaining the precise geometry of the functional motif and ensuring the overall stability of the scaffold structure. The assessment must consider both the structural accuracy of the motif placement and the broader protein context that supports it, making this a multi-scale evaluation challenge.

We measure motif scaffolding in terms of both motif accuracy and overall designability. For motif accuracy, we calculate RMSD between the input motif structure and the corresponding region of the designed protein to assess whether the motif structure is preserved (motifRMSD $<1.0$). As for the overall designability, we use scTM score $>0.8$ as being designable. We have accordingly elaborated on motif-scaffolding evaluation in the appendix.

[**Datasets**] Datasets typically include libraries of known functional motifs (e.g., catalytic sites, binding interfaces) and diverse scaffold structures that can potentially accommodate these motifs. The Protein Data Bank is a primary source, but curated datasets of functional sites like the Catalytic Site Atlas are also valuable.

[**Related benchmarks**] Enzyme Design Challenge provides relevant test cases. However, given the specificity of motif scaffolding tasks, benchmarks often need to be tailored to the particular class of motifs or functions being targeted. Currently, there exists no comprehensive benchmark for this task in the field. A widely used benchmark containing 17 (25) motif-scaffolding problems was used in RFDiffusion (Watson et al., 2023b).

[**Evaluation Metrics**] Following Yim et al. (2024), we implement several key metrics.

- **Motif Accuracy** We measure the structural retention of motif placement using RMSD calculations, focusing on the geometric alignment of critical functional elements.

- **Overall Designability** We assess the overall stability and structural integrity of the designed protein framework using metrics using self-consistency TM Score.

Figure 3: Sequence-based, structure-based, and co-generation evaluation pipeline of motif-scaffolding.

| | sequence-based | |
|---|---|---|
| prediction | seq$_{pred}$: ✓ | struct$_{pred}$: ✗ |
| motif-preserving | RMSD(ESMFold(seq$_{pred}$)[motif],struct$_{native}$[motif])<1.0 | |
| designability | pLDDT(ESMFold(seq$_{pred}$))>70 | |

| | structure-based | |
|---|---|---|
| prediction | seq$_{pred}$: ✗ | struct$_{pred}$: ✓ |
| motif-preserving | RMSD(ESMFold(PMPNN(struct$_{pred}$))[motif],struct$_{native}$[motif])<1.0 | |
| designability | TMScore(ESMFold(PMPNN(struct$_{pred}$)), struct$_{pred}$)>0.8 | |

| | co-generation | |
|---|---|---|
| prediction | seq$_{pred}$: ✓ | struct$_{pred}$: ✓ |
| motif-preserving | RMSD(ESMFold(seq$_{pred}$)[motif],struct$_{native}$[motif])<1.0 | |
| designability | TMScore(ESMFold(seq$_{pred}$), struct$_{pred}$)>0.8 | |

**[Model implementations]** Structure-based models (FrameFlow, RFDiffusion), sequence-based models (DPLM, EvoDiff), and multimodal models (ESM3) require different ways to take as input the motif information and generate the scaffolds. For example, structure-based methods require an extra sequence design model to predict sequences of the designed proteins, while sequence-based methods cannot directly read motif structure and also require an extra folding model to predict the structure of the designed proteins. Multimodal approaches, on the other hand, can read and predict structure and sequence simultaneously by themselves. Hence, comparing them in completely identical settings is challenging, and we resort to slightly different criteria to evaluate these approaches.

We focus on two aspects to assess the success of motif scaffolding: overall designability and motif-preservation. The overall illustration is shown in Figure 3. Specifically, (1) For the sequence-based method, we only take the generated sequence and utilize ESMFold to obtain the predicted structure, and the pLDDT score provided by ESMFold is used to assess overall quality. (2) For the structure-based method, we only take the generated structure and then leverage ProteinMPNN to predict the sequence, followed by ESMFold to predict the structure, where overall quality is assessed by scTM. (3) For the co-generation method, we take both the generated structure and sequence and predict the structure given the generated sequence with ESMFold, where scTM is calculated between the generated structure and ESMFold predicted structure to evaluate overall quality. Considering that the ground truth motif structure is given, we only utilize the ESMFold predicted structure to calculate motif-RMSD.

### B.1.6 Antibody Design

**[Task Definition]** The goal of antibody design is to generate antibodies that can specifically bind to a given antigen. Since the Complementarity-Determining Regions (CDRs) of antibodies are highly variable and primarily responsible for antigen binding, antibody design could be simplified to the design of CDR regions and further reduced to the design of the third CDR in heavy chain (CDR-H3). Given the crucial role that protein structure plays in interactions, antibody design usually involves the simultaneous design of the sequence and the structure when binding to the antigen.

**[Evaluation Metrics]** As mentioned in the main text, antibody design can ultimately be simplified to the design of CDR-H3. Therefore, in this study, we evaluate the performance of different antibody design methods by evaluating the CDR-H3 sequences generated by these methods. Given the primary objective of this study is to assess the relative performance of various design models rather than the in vivo/vitro functionality of the antibodies they generate, we opted to directly evaluate the designed antibodies using their predicted structures. This approach is grounded in several considerations: firstly, it ensures a clear focus on evaluating the design methodology itself, independent of experimental constraints. Secondly, the significant time and resources required for extensive experimental validations, as well as the limitations of methods that can accurately simulate the real binding structure of antibodies, render in vivo/vitro assessments impractical. Direct evaluation of

the designed structures presents a feasible and efficient strategy that aligns with the study's goals and resource constraints while still providing valuable theoretical benchmarks for subsequent experimental investigations.

For methods capable of generating multiple antibodies for the same antigen, we generated 64 CDR-H3 sequences per antigen using each method and calculated the average performance across these different generated samples. Additionally, we also calculated the standard deviation of the performance among different samples generated for a single antigen.

As a highly goal-oriented functional protein design task, the evaluation of antibody design is straightforward, namely the **Functionality** (binding capability to the target antigen) and **Specificity** of the designed antibody. Additionally, the **Rationality** of the designed antibodies sequence and structure needs to be evaluated for filtering out invalid designs. However, it remains a challenging task to accurately simulate the performance of antibodies in real wet-lab experiments, including functionality and specificity, using computational methods, and there is a lack of reliable approaches. Therefore, we focus on evaluating antibody design models. Superior models can often provide a better understanding of interactions between antibodies and antigens, potentially leading to the production of improved antibodies. Although both the evaluation of design models and antibodies involve assessing antibodies, their approaches differ. The former directly evaluates the structures and sequences generated by the models without modifications, whereas the latter focuses on the actual performance of antibodies in experiments, meaning that their structures may change and differ from the design. Existing studies also evaluate the **Accuracy** of designed antibodies by measuring their similarity to natural antibodies as natural ones are confirmed to be effective. However, using accuracy as an evaluation metric is inadequate in many cases, we demonstrate the misleadingness of AAR and RMSD in 2.1.6.

**Accuracy**:

- AAR: AAR is the accuracy evaluation of generated sequences compared to reference/natural sequences. For the calculation of AAR (Amino Acid Recovery Rate), similar to existing work, we calculated the number of residues in the generated CDR-H3 sequences that match the natural antibody.

- RMSD: RMSD is the consistency evaluation of generated structures compared to reference/natural structures. In the calculation of RMSD (Root Mean Square Deviation), we measured the RMSD of the generated and natural antibodies in the CA coordinates of the CDR-H3 region. For methods other than dyMEAN, since their task setting provides the true binding pose of the antibody FR region and antigen, there is no need to align the generated structure with the natural structure when calculating RMSD. For dyMEAN, we aligned the 2 FR residues at each end of CDR-H3 with the corresponding residues in the natural antibody, applied the obtained transformation to CDR-H3, and then calculated the RMSD.

- TM-score: TM-score is also the consistency evaluation of generated structures compared to reference/natural structures. We calculated the TM-score only for the CDR-H3 region. To this end, we saved the generated CDR-H3 part as a .pdb file and used `TMalign` (Zhang & Skolnick, 2005) to calculate the TM-score between the generated CDR-H3 and the natural CDR-H3.

**Functionality**:

- Binding Energy: Binding Energy indicates the strength of antibody-antigen binding with the generated structures and we use Rosetta to calculate the Binding Energy. The calculation of binding energy requires the all-atom structure of the protein, while most methods only generate the backbone atom structure. Therefore, we first used Rosetta to pack the missing side-chain atoms. Subsequently, we optimized the side-chains in the CDR-H3 region using Rosetta minimization while keeping the backbone structure unchanged to ensure that the CDR-H3 generated by the model reaches the minimum energy state in the binding environment with the antigen. During minimization, we set the step to 100 (we tried using more steps and repeats, although the energy did further decrease, the reduction was very limited and much smaller than the energy difference between different methods; however, the time consumption significantly increased). After minimization, we calculated the energy on the all-atom structure. Finally, we used the `InterfaceAnalyzer` in Rosetta to calculate the binding energy between CDR-H3 and the antigen.

**Specificity**:

- SeqSim: We use SeqSim to detect the mode collapse in sequence generation (or sequence specificity towards different antigens), which indicates that the generated antibodies lost the specificity for specific antigens. SeqSim is defined as the average similarity between any sequence pairs among the generated sequences. First, we introduce the definition and implementation of similarity. The similarity between two sequences is defined as the percentage of matched amino acids over the aligned length after alignment (thus, this metric is affected by the length gap between the two sequences). Given that our goal is to calculate the number of matches rather than the matching score and that the two ends of CDR-H3 are fixed to FR3 and FR4, we need an alignment method that: (1) assigns a score of 1 for matches, and 0 for gaps and mismatches; (2) does not introduce gaps at the two ends of CDR-H3. We used the `PairwiseAligner` in Biopython (Cock et al., 2009) for sequence alignment, setting `match_score` to 1, all other scores to 0, and the `end_gap_score` to *-inf* so that the alignment process meets our requirements. For methods that generate only one antibody per antigen, we directly calculate the average SeqSim among the 55 generated CDR-H3 sequences as SeqSim-outer. For methods that generate multiple antibodies, we calculate the average SeqSim between two sets of sequences generated for two antigens as SeqSim-outer and also calculate the average SeqSim within each set as SeqSim-inner. The formulas for calculating SeqSim-outer and SeqSim-inner are as follows:

$$\text{SeqSim-outer} = \frac{1}{N*(N-1)*M^2} \sum_{i=1}^{N} \sum_{j=1|j\neq i}^{N} \sum_{x=1}^{M} \sum_{y=1}^{M} \text{SeqSim}(s_i^x, s_j^y), \qquad (1)$$

$$\text{SeqSim-inner} = \frac{1}{N*M*(M-1)} \sum_{i=1}^{N} \sum_{x=1}^{M} \sum_{y=1|y\neq x}^{M} \text{SeqSim}(s_i^x, s_i^y), \qquad (2)$$

where $N$ denotes the number of antigens in the test set ($N$=55 in this study), $M$ denotes the number of samples generated for each antigen ($M$=64 in this study), and $s_i^x$ represents the $x$-th CDR-H3 sequence generated for the $i$-th antigen.

- PHR: PHR is the proportion of hydrophobic residues in the generated CDR-H3 sequences, can also reflect the lack of specificity, as the binding caused by the interactions generated by these residues is generally considered to lack antigen specificity. Although both PHR and SeqSim are used to represent the specificity of antibody design methods, they focus on different aspects. Thus, the same method may exhibit different tendencies in these two metrics (SeqSim can be understood as an evaluation of the method's specificity, while PHR is an evaluation of the specificity of the generated antibodies. When SeqSim performs poorly, the performance of PHR is of limited significance). For example, AbDPO achieves high SeqSim-outer but does not perform well in PHR. This indicates that AbDPO can specifically design antibodies for different antigens, but these antibodies contain many hydrophobic residues, leading to potential nonspecific interactions with multiple proteins.

**Rationality**:

- CN-Score: CN-Score is the evaluation of the rationality of the structure by scoring the distribution of generated peptide bond length. To evaluate the consistency of the peptide bond length of generated antibodies with that of natural antibodies, we fit a Kernel Density Estimation (KDE) function using the length of peptide bonds found within the CDR-H3 regions of natural antibodies. The density of the generated peptide bond length, CN-Score, is used to represent the consistency. For generated peptide bonds shorter than the minimum natural peptide bond length or longer than the maximum, the density is defined as 0. The final CN-Score for a generated antibody is defined as the average density of the lengths of all its peptide bonds. It is important to note that the length variation of peptide bonds is very small, which leads to a very narrow distribution of natural peptide bond lengths. When the generated peptide bond length deviates slightly from the average length (1.3310), its density in the KDE function will sharply decrease, which explains why all methods show a significant difference in CN-Score compared to natural antibodies.

- Clashes: Clashes is the assessment of the potential clashes. Although atomic clashes within proteins mainly occur between the side chains, most methods do not generate the side chains of residues. Using packing methods to complete side chains can always find a side-chain conformation with the fewest clashes through extensive searching. Therefore, we instead evaluate the

potential clash level in the generated structures rather than the specific number of clashes. To do this, we calculate the CA distance between two residues; when the CA-CA distance between two residues not connected by a covalent bond is less than the minimum CA-CA distance commonly found in covalently bonded residues (3.6574, derived from the CA-CA distance statistics in the CDR-H3 region of the RAbD dataset), we consider these two residues to have potential clashes. We then calculate the number of residue pairs with distances below this threshold to measure the clash level in the generated structures. The difference between Clashes-inner and Clashes-outer is: Clashes-inner measures the clash level within the generated CDR-H3 structure, while Clashes-outer measures the clash level between the generated CDR-H3 structure and other components, including the antigen, the heavy chain FR region, and the light chain of the antibody.

- SeqNat: SeqNat, Sequence naturalness, is the evaluation of the rationality of the generated sequence. To measure how close the designed CDR-H3 sequences are to natural sequences, we used the pLL predicted by the AntiBERTy model. We input the entire heavy chain sequence into the model, which means that AntiBERTy makes predictions based on the entire heavy chain of the antibody, but unlike the standard procedure in AntiBERTy, the pLL calculation area is only within the CDR-H3 region (the standard procedure calculates pLL over the entire input sequence).

- Total Energy: Total Energy is the evaluation of the joint rationality of the generated sequence and structure from the perspective of physical energy. Before calculating the total energy, we performed the same energy optimization process on the designed CDR-H3 regions as described in the Functionality section. We then used Rosetta's full atom score function with the default weights from REF15 (Alford et al., 2017) to calculate the total energy of each residue in the CDR-H3 region. The Total Energy of the CDR-H3 region is defined as the sum of the total energy of all its residues.

- scRMSD: We use scRMSD to evaluate the model's ability of structural modeling by calculating the difference between the designed structure and the simulated structure. In this metric, we used a two-stage method to predict the structure of the generated sequences. In the first stage, we used IgFold to predict the structure based on the sequence pair of the antibody's light and heavy chains (although the region we evaluate only exists in the heavy chain, and IgFold also supports single-chain input, we found that inputting two chains results in higher accuracy). The real structure of the non-CDR-H3 regions of the antibody was also provided as a template to obtain the initial predicted structure. We then used the Kabsch algorithm to align the non-CDR-H3 regions of the heavy chain with the real structure and applied the resulting transformation to the predicted CDR-H3 structure. This aligns the predicted CDR-H3 structure to its original complex. At this point, the CA-RMSD between the predicted CDR-H3 structure and the real structure in the RAbD dataset is 1.95. The structure predicted by IgFold is unrelated to the antigen, and since the antibody undergoes conformational changes in the binding interface after binding with the antigen, we used Rosetta to relax the predicted CDR-H3 in the presence of the antigen in the second stage. The relaxation involves changes in both the backbone and side-chain structures. Specifically, we repeated relaxation runs five times for each structure predicted by IgFold, with 200 steps each time, and selected the structure with the lowest energy as the final predicted structure. At this stage, the CA-RMSD with the real structure decreased to 1.77. We then calculated the RMSD of the CA coordinates between the predicted structure and the backbone CA coordinates generated by the model, which is referred to as scRMSD.

**[Datasets]** The Structural Antibody Database (SAbDab Dunbar et al. (2013)) is the commonly used dataset for antibody design. It contains structural data of the antibody-antigen complex, but the data size is limited and contains numerous redundancies. Although SAbDab's official statistics indicate that the database includes over 8,000 entries of complexes containing antigens, only more than 3,000 entries remain after deduplication. Furthermore, researchers typically cluster the SAbDab data based on the sequence identity of CDR-H3, with a clustering threshold generally set at 40% identity. Subsequently, within different clusters, the data is divided into training, validation, and test sets.

**[Model Implementations]** We retrained all the methods with unified training data and the official training config for a fair comparison and evaluated the methods with unified test data.

**Training data**:
To build the unified training data, we use antibody-antigen complex structural data from the SAbDab dataset under the IMGT scheme (Lefranc et al., 2009) as the training dataset. We collected

antigen-antibody complexes with both heavy and light chains and protein antigens. We then discarded duplicate data with the same CDR-L3 and CDR-H3 sequence. The remaining complexes are used to cluster via MMseqs2 (Steinegger & Söding, 2017) with 40% sequence similarity as the threshold based on the CDR-H3 sequence of each complex. Finally, we select the clusters that do not contain complexes in the RAbD dataset and split the complexes into training and validation sets with a ratio of 9:1 (1786 and 193 complexes respectively).

**Test data**:
To build the unified test data, we extracted 55 antibody-antigen complexes from the RAbD dataset. The original RAbD dataset contains 60 antibody-antigen complexes. In this study, we hope that the evaluation of antibody design methods is based on antibodies that contain both light and heavy chains, and simultaneously the antigen contains at least one protein chain. In practice, **2ghw** and **3uzq** lack light chains, while **3h3b** lack heavy chains. **5d96** is excluded because of the incorrect chain ID information in rabd_summary.jsonl[4], where heavy chain $J$ and light chain $I$ do not bind to antigen chain $A$. **4etq** is excluded as HERN reported an error when running for this complex.

**Model**:
All the models are retrained with their default training config. It should be noted that **[dyMEAN-FixFR]** is not an official variant of dyMEAN, and we implemented this variant for a fair comparison with other methods. Unlike other methods, which are designed to accept the true structure of the antibody-antigen complex and generate the missing CDR-H3 region, dyMEAN is set up to accept only the structure of the antigen and the sequence of the non-CDR-H3 regions of the antibody. Therefore, the model needs to both generate the CDR-H3 region and predict the overall structure of the antibody as well as the binding pose between the antibody and antigen. Incorrect pose estimation can severely affect the interactions between CDR-H3 and the antigen, making a direct comparison between dyMEAN and other methods unfair. To compare dyMEAN with other methods more fairly, we made some modifications to dyMEAN by providing the true structure of the non-CDR-H3 regions of the antibody and the binding pose, aligning dyMEAN with the other methods. In dyMEAN-FixFR, we also used Rosetta (Alford et al., 2017) to repack the side chains, consistent with other methods, to avoid the influence of the side chains generated by dyMEAN on the evaluation results. Additionally, we introduced some randomness in the initialization of the structure, which allows dyMEAN-FixFR to generate multiple different antibodies for the same antigen.

**[Extended Explanations and Discussion on Model Performance]**

**Specificity**:

- In **SeqSim-outer**, we noted that MEAN and dyMEAN generated highly similar sequences for different antigens (the maximum **SeqSim-outer** in our test set was 0.79, indicating that all antibody differences came only from length variations). This suggests that their excellent AAR might stem from learning high-frequency patterns in antibody sequences, generating antibodies according to these patterns for different antigens. In contrast, DiffAb and AbDPO performed the best.

- For methods that can generate different antibodies for the same antigen, we also measured the sequence similarity among different antibodies generated for the same antigen (**SeqSim-inner**). We expect antibodies generated for the same antigen to be more similar. In this aspect, dyMEAN-FixFR and AbDPO performed the best. However, the 0.96 **SeqSim-inner** of dyMEAN-FixFR indicates that despite introducing randomness during model initialization, the final sequence generation showed almost no differences. Additionally, DiffAb, which performed best in **SeqSim-outer**, generated less similar antibodies for the same antigen, suggesting possible underfitting in sequence generation. Considering both types of **SeqSim**, AbDPO achieved the best performance.

- In **PHR**, HERN and dyMEAN performed the best, but overall, almost all methods performed better than natural antibodies. Only AbDPO generated an excessive number of hydrophobic residues, reducing specificity. However, its variant, AbDPO++, controlled **PHR** well, closely matching natural antibodies among all methods.

**Rationality**:

---

[4]https://github.com/THUNLP-MT/MEAN/blob/main/summaries/rabd_summary.jsonl

- In structural rationality, we focused on the score for peptide bond lengths conforming to the natural peptide bonds length distribution (**CN-score**), the number of potential internal clashes in the generated structure (**Clashes-inner**), and the clashes between the generated structure and other parts (**Clashes-outer**). It was evident that irrational structures were prevalent in generated antibodies, but overall, diffusion-based methods performed better. AbDPO++ and DiffAb achieved the best performance among all methods. HERN and MEAN/dyMEAN exhibited different tendencies in **Clashes-inner/outer**, corresponding to our observations of the generated samples. HERN tends to generate large CDR-H3 structures, leading to fewer internal clashes but more clashes with the antigen, whereas MEAN/dyMEAN tends to generate smaller CDR-H3 structures.

- In sequence rationality, we used the inverse perplexity of AntiBERTy (Ruffolo et al., 2021) to represent sequence naturalness, **SeqNat**, showing that HERN performed the best, possibly due to HERN being the only auto-regressive model. AbDPO++ achieved the second-best performance and was closest to natural antibodies.

- In the joint evaluation of structure and sequence, we mainly focused on the consistency between the generated structure and sequence from two perspectives: physical energy and structure prediction. In terms of physical energy, we calculated the total energy of the generated CDR-H3s (**Total Energy**), which would be severely affected by the clashes caused by sidechains and thus reflect the irrationality between the generated structure and sequence. In this energy-related metric, AbDPO and AbDPO++ performed best among all methods. From the perspective of structure prediction, we used IgFold (Ruffolo et al., 2023) to predict the structure of the generated sequence, performed a post-optimization with the antigen as the condition, and calculated the CA-RMSD between the predicted structure and the generated structure (**scRMSD**). dyMEAN and dyMEAN-FixFR performed best in **scRMSD**. Although these two metrics both reflect the consistency between sequence and structure, they focus on different aspects. Moreover, both energy calculations and structure predictions have inherent errors, so the performance of different methods may not be consistent across these two metrics.

### B.2 Protein Conformation Prediction

#### B.2.1 Protein Folding: single-state prediction

**[Task Definition]** Protein folding task predicts the 3D structure of a protein based on its sequence. Folding models such as AlphaFold2 (Jumper et al., 2021) have achieved unprecedented accuracy in predicting protein structures at scale, complementing experimental characterizations and driving advancements in biology and drug discovery (Varadi et al., 2022). From a modeling perspective, sequence-to-structure prediction is a critical measure of a model's understanding of these two modalities. The ability to translate sequences into structures forms the foundation for recent progress in protein conformation prediction (Jing et al., 2024; Zheng et al., 2024; Wang et al., 2024c). As such, we recognize the necessity of including protein folding in this benchmark, viewing it as a specific instance of protein conformation prediction for a single conformational state.

**[Evaluation Metrics]** The primary goal of evaluating protein folding models is to assess their *accuracy* in predicting structures of unseen proteins. This is done by comparing the predicted structures to reference structures, such as experimentally determined ones available in the Protein Data Bank (PDB). To ensure an unbiased evaluation, time-based splits are commonly employed, using recently deposited structures of previously unseen proteins for benchmarking (cas, 2022; Robin et al., 2021). Beyond accuracy, the ability to predict protein structures with minimal structural violations provides a reference-free measure of a model's capability to generate high-**quality** protein conformations. Unlike the design tasks discussed earlier, structural diversity is not a focus in this evaluation. Detailed implementations are provided below:

- **Accuracy:** Structural accuracy is measured by the structural similarity with reference structures. Specifically, *global* similarity metrics including TM-score, RMSD and global distance test (GDT) are calculated using `TMscore` Zhang & Skolnick (2004) obtained from `https://zhanggroup.org/TM-score/`. We use `-seq` option to align sequences before structural alignment. Local distance difference test (lDDT) is an alignment-free method to compare *local* structural similarity. We calculate the value using the original implementation (Mariani et al., 2013) from `https://swissmodel.expasy.org/lddt/downloads/`.

- **Quality:** The structural quality of generated conformations are assessed by CA clash % and PepBond break %:

- CA clash % is the rate of potential residue-residue clashes based on the positions of alpha-carbon atoms. A *clash* is determined if the distance between a pair of alpha carbon atoms is less than 3.0 Å, similar to Lu et al. (2024). And CA clash % is calculated as

$$\text{CA clash \%} = \frac{\text{number of residues with clashes}}{\text{sequence length}} \times 100\%.$$

- PepBond break % evaluates the potential peptide bond (C-N) break between connecting residues, providing a more rigorous metric about inter-residue disconnection than CA level metrics used in Lu et al. (2024). We use a maximum peptide length threshold of 1.4 Å to determine a chain break, as suggested by the Biopython implementation [5]. Similarly, PepBond break % is calculated as

$$\text{PepBond break \%} = \frac{\text{number of C-N bond break}}{\text{sequence length} - 1} \times 100\%.$$

**[Datasets]** The folding models included in this benchmark are those that serve as base models for protein conformational predictions and were established prior to 2022. We use CAMEO2022 from Jing et al. (2023) for evaluation, which consists of 183 short-to-mid-length single protein chains ($<$ 750 amino acids) from the targets of CAMEO (a continuous benchmarking initiative for structure prediction of newly deposited protein structures) between Aug 1 and Oct 31, 2022. CAMEO2022 consists of 183 single protein chains collected from CAMEO targets between August and October 2022, with sequence lengths of less than 750 amino acids, following Jing et al. (2023). Protein sequences and structures were extracted from the mmCIF files available at the RCSB Protein Data Bank (https://www.rcsb.org/, Berman et al. (2000)). One of the proteins (PDB ID: 8AHP, chain A) has since been superseded by a new PDB entry 8QCW and we have replaced this chain with the updated record.

**[Model Implementations]** Several folding models, including AlphaFold2, OpenFold, RoseTTAFold2, use Multiple Sequence Alignment (MSA) as sequence input. We standardize MSA curation using the querying pipeline and the online server provided by ColabFold (Mirdita et al., 2022). Templates are not provided in model inference. Additional model implementation details for each model are as follows:

- AlphaFold2 (Jumper et al., 2021): We used the ColabFold implementation Mirdita et al. (2022) for AlphaFold2 inference. All five models (with pTM) were used to predict five candidate structures, and the structure with the highest pLDDT confidence score was selected for performance evaluation. All models were run with default settings.

- OpenFold (Ahdritz et al., 2022): We used `openfold v2.0.0` for inference with their pretrained OpenFold weights (with pTM). Since only one checkpoint (`finetuning_no_templ_ptm_1`) corresponding to the model configuration `model_3_ptm` is available, we generated three structures using three random seeds and made a total of 5 predictions. The structure with the highest pLDDT score was selected for performance evaluation.

- ESMFold (Lin et al., 2023a): We use the public ESM repository for inference with the model `esm.pretrained.esmfold_v1`. Since EMSFold predictions are deterministic, we generated only one structure per protein for performance evaluation.

- RoseTTAFold2 (Baek et al., 2023): We follow their official repository and instructions for inference. Only one structure per protein was predicted for performance evaluation.

- EigenFold (Jing et al., 2023): We follow the official repository, weights, and the setups provided by the authors for inference. In the *protein folding* task, we sampled 5 structures for each protein and selected the one with the highest ELBO estimation for performance evaluation. Because EigenFold can not predict sequences containing unknown amino acids (labeled 'X'), we removed the 'X' in the input sequences, as done in the original implementation.

**[Extended Explanations and Discussion on Model Performance]**
Our benchmarking results (Table 7 align with previous reports (Jing et al., 2023), showing that MSA-based folding model generally outperforms protein-language-model-based folding model. Eigen-Fold (Jing et al., 2023), one of the first diffusion generative models for both protein folding and

---

[5] https://biopython.org/docs/dev/api/Bio.PDB.internal_coords.html#Bio.PDB.internal_coords.IC_Chain

conformation prediction, shows relatively weaker performance on the folding task. Its performance could be limited by several design factors: it is built on OmegaFold Wu et al. (2022), uses a coarse-grained representation with only alpha carbons, and has a small model size of 572K trainable parameters.

### B.2.2 MULTIPLE-STATE PREDICTION

**[Task Definition]** Multiple-state prediction aims to accurately generate two or more distinct conformational states of a protein. These states are typically associated with functional conformational changes, such as those induced by ligand binding, or metastable states observed during molecular dynamics simulations. The ability to predict these "alternative" conformations, in addition to the folded structure, offers valuable insights into a model's capability to generate plausible stable conformational states. This serves as an essential first step toward understanding protein conformational dynamics.

**[Evaluation Metrics]** The **accuracy** and **quality** metrics for multiple-state prediction are naturally derived from those used in single-state prediction, with some modifications. Unlike single-state prediction, multiple-state prediction involves sampling an ensemble of conformations from the model. Given a fixed sample size, the accuracy of recovering one state can be evaluated as the best accuracy among all samples compared to the reference structure. For direct model comparison, a single accuracy score is preferred to represent the average performance across the recovery of different states. Additionally, transitioning from folding to conformation generation introduces the need to evaluate the **diversity** of the generated samples, reflecting the model's ability to capture a range of plausible conformational states. Implementation details are as follows:

- **Accuracy:** The accuracy of predicting a conformational state is determined by the best structural similarity among the samples to the reference structure, measured by TM-score (for *apo-holo*) or RMSD (for BPTI). Use RMSD as an example:

$$\text{Accuracy of state } k = \min_{\mathbf{x}_i \in \text{samples}} \text{RMSD}(\mathbf{x}_i, \mathbf{x}_k^{\text{ref}})$$

where $\mathbf{x}_k^{\text{ref}}$ is the reference structure for state $k$. We then calculate the average accuracy across states as in Jing et al. (2023), referred as "ensemble accuracy":

$$\text{Ensemble RMSD} = \frac{1}{K} \sum_{k \in K \text{ states}} \text{Accuracy of state } k$$

- **Diversity:** Diversity is evaluated by the average pairwise structural similarity among the generated samples for a protein, measured using TM-score or RMSD. To reduce computation time, we randomly sample 100 pairs of samples for estimation.
- **Quality:** The structural quality is evaluated using CA clash and PepBond break, see single-state prediction for details.

**[Datasets]** We benchmark the models on two datasets reflect common scenarios in the study of protein conformational changes: *apo-holo* captures the conformational changes related to specific protein function (i.e., ligand-binding processes) (Saldaño et al., 2022) and BPTI captures the metastable states discovered from long-time MD simulations (Shaw et al., 2010). Specifically:

- *Apo-holo* consists of 91 single chain proteins curated by Saldaño et al. (2022), each featuring a pair of experimentally determined conformations: *apo* (unbound) and *holo* (bound), representing a two-state prediction task related to ligand-binding. The protein sequences and the structures of both *apo* and *holo* conformations were extracted using the same pipeline as in CAMEO2022. Consistent with Jing et al. (2023), we use the sequences of the *apo* state as the primary sequence for model inference. A total of 20 conformations are sampled for each protein during evaluation.
- **BPTI** (Bovine Pancreatic Trypsin Inhibitor) is a 58-amino-acid protein whose dynamics have been extensively studied through long-time MD simulations (Shaw et al., 2010). We use the structures of the five cluster centers identified in the MD study as reference structures. This represents a five-state prediction task with Cluster 3 being the most challenging to sample (Wang et al., 2024c). A total of 1,000 conformations are sampled for evaluation.

**[Model implementations]**

- EigenFold (Jing et al., 2023): The implementation is the same as in the folding task. See the section above for details.

- MSA-subsampling (Del Alamo et al., 2022): We implemented MSA-subsampling using the `openfold v2.0.0` package by adjusting the two configuration parameters, `max_msa_clusters` and `max_extra_msa`, following Del Alamo et al. (2022). Specifically, we refer to `max_extra_msa` as the MSA depth and set `max_msa_clusters` to half that depth, while keeping other OpenFold settings at their default values. The original MSAs were obtained using the same ColabFold pipeline as in AlphaFold2.

- Str2Str (Lu et al., 2024): We followed the official implementation of Str2Str and used OpenFold-predicted structure as the initial structures. Ensemble results were collected by uniformly sampling from $t$ values. For BPTI, we used the author-recommended noising schedule with maximum forward time of $T_{\max} = 0.15$ ($t = 0.10, 0.15$). For *apo-holo* and ATLAS datasets, we experimented with $T_{\max} = 0.14$ ($t = 0.06, 0.08, 0.10, 0.12, 0.14$) and $T_{\max} = 0.3$ ($t = 0.06, 0.12, 0.18, 0.24, 0.30$) for both the SDE and ODE models.

- AlphaFlow/ESMFlow (Jing et al., 2024): We used the official repository and released model weights for inference. The MSAs for AlphaFlow models were obtained through ColabFold's pipeline. We included models pretrained on PDB (-PDB) and fine-tuned on MD datasets (-MD).

- ConfDiff (Wang et al., 2024c): We followed the authors' implementation and used the released weights for inference. In this benchmark, we used recycle3 representations for both ConfDiff-Open and ConfDiff-ESM models, with comparison between classifier-free guidance models (-ClsFree), PDB base models (-PDB), and MD data fine-tuned models (-MD). The energy and force guidance models are dataset-specific and are only available for the BPTI dataset with ESMFold representations.

**[Extended comparison on protein conformation models]**

Recent works on protein conformational prediction have explored several strategies to extend folding models to generate multiple conformations. A common goal across these studies is to enhance sample diversity while ensuring that the generated conformations remain accurate and faithful to the protein, given the high-dimensional space of protein structure. Below, we briefly highlight the key differences among the studies and strategies evaluated in this benchmark:

- **Perturbing the input of folding models.** While AlphaFold2 is designed to predict a single folded structure of a protein, several studies have proposed perturbing its MSA input to generate alternative structures (as a proxy to conformations) without re-training the model (Del Alamo et al., 2022; Wayment-Steele et al., 2024). In this benchmark, we assess MSA subsampling, a method that reduces the number of input MSAs (referred to as "depth") by subsampling the full MSA, enabling the prediction of different structures due to the depletion of the input information. The depth of MSA controls the trade-off between the sample diversity and how faithful the structure is to the protein.

- **Perturbing folded structures.** Instead of perturbing the input to a folding model, Str2Str (Lu et al., 2024) perturbs the structure predicted from a folding model. It uses a structure-only diffusion model (i.e., a backbone design model) to perturb the input structure through a forward-backward diffusion process. The level of perturbation is controlled by the maximum diffusion time, $T_{\max}$. They also used ensembling by sampling at various diffusion times $t \leq T_{\max}$.

- **Training generative models on large-scale structural data from experiments or simulations.** A more direct approach involves training sequence-conditioned generative models using diffusion or flow frameworks. EigenFold (Jing et al., 2023), AlphaFlow (Jing et al., 2024), and ConfDiff (Wang et al., 2024c) follow similar approaches by fine-tuning a diffusion time $t$-dependent score or denoising model based on folding models, using structural data from PDB. Specifically, AlphaFlow finetunes all layers of AlphaFold2, while EigenFold and ConfDiff use pretrained representations from folding models and train a lightweight add-on module for score or denoising prediction. Despite adopting a generative framework, models solely trained on PDB data are limited in predicting conformational distributions. To address this, AlphaFlow and ConfDiff further fine-tuned their models on a recent MD dataset containing densely sampled conformations for proteins (see Atlas in the Datasets section).

- **Integrate physical priors in conformational training or sampling.** Due to limited availability of large-scale protein conformation data from MD simulation, some models have explored inte-

grating structural and physical priors during training. ConfDiff (Wang et al., 2024c) introduced two guidance techniques to improve conformational sampling: (1) classifier-free guidance, which combines a sequence-conditioned conformation model with an unconditional (structure-only) model to explore conformational space (ConfDiff-ClsFree), and (2) energy/force guidance, which directs sampling toward regions with lower potential energy (ConfDiff-Energy/Force) through auxiliary prediction modules for intermediate energy/force guidance. However, such physical prediction modules are dataset-specific and requires training additional modules.

**[Extended Explanations and Discussion on Model Performance]**

The complete evaluation results for multiple-state prediction (BPTI and apo-holo) are shown in Table 13 and Table 14.

For BPTI, as discussed in the main text, certain conformation exploration techniques, such as MSA subsampling and guidance used in ConfDiff, have shown their ability to sample diverse structures while staying faithful to the protein. In contast, structure-only approaches like Str2Str perform poorly on this task, likely because these models do not ensure that the perturbed structure remains faithful to the provided sequence. On the other hand, EigenFold shows limited diversity, as it was trained solely on PDB structures and does not incorperate conformation exploration strategies. This limits its effectiveness in sampling diverse samples. While AlphaFlow and ESMFlow demonstrate competitive performance, fine-tuning on the MD dataset introduces trade-offs in quality, notably an increased incidence of peptide bond breaking between residues.

For *apo-holo*, strategies to improve sample diversity – such as reducing MSA depth, applying structural perturbation, fine-tuning on MD conformation data, or using classifier-free guidance – generally do not improve (and sometimes even reduce) the TMens score. Interestingly, we found that the best-performing models are those that most closely resemble folding models (e.g., MSA-depth256, AlphaFlow-PDB). The included baseline *apo*, that always predicts the perfect *apo* structures, confirmed that a higher TMens score can result from accurate prediction of one of the states. These findings suggest that using better folding model provide a strong baseline performance for conformational sampling but none of the current models show clear evidence of effectively modeling conformational changes during complex biological processes, such as ligand binding.

Table 13: Complete performance on the multiple-state prediction of BPTI. Accuracy metrics (RMSDens, RMSD Cluster 3) are reported as the mean and standard deviations from 20 bootstrap samples with replacement, at different sample sizes ($N = 10 \sim 1000$). Diversity and Quality scores are evaluated based on 1,000 conformations for each model. The **best** performance is highlighted in bold, and the second-best is underlined. "N/A" indicates not applicable due to model resolution. RMSD is measured in Å.

| | RMSDens ↓ | | | RMSD Cluster 3 ↓ | | | Diversity | Quality | |
|---|---|---|---|---|---|---|---|---|---|
| | N=10 | N=100 | N=1000 | N=10 | N=100 | N=1000 | Pairwise RMSD | CA clash% ↓ | PepBond break%↓ |
| EigenFold | 1.56±0.02 | 1.50±0.01 | 1.46±0.00 | 2.54±0.03 | 2.48±0.01 | 2.46±0.01 | 0.85 | 1.4 | N/A |
| MSA-depth256 | 1.58±0.01 | 1.54±0.01 | 1.52±0.01 | 2.51±0.02 | 2.48±0.01 | 2.44±0.01 | 0.20 | **0.0** | 9.2 |
| MSA-depth64 | 1.60±0.01 | 1.55±0.02 | 1.51±0.01 | 2.46±0.03 | 2.41±0.04 | 2.34±0.03 | 0.55 | **0.0** | 7.9 |
| MSA-depth32 | 1.66±0.03 | 1.54±0.04 | 1.41±0.02 | **2.43±0.06** | **2.19±0.16** | **1.85±0.05** | 2.14 | 0.6 | 10.6 |
| Str2Str-ODE ($T_{max} = 0.15$) | 2.40±0.12 | 2.20±0.05 | 2.09±0.01 | 3.00±0.20 | 2.73±0.12 | 2.58±0.05 | 1.86 | **0.0** | 13.9 |
| Str2Str-SDE ($T_{max} = 0.15$) | 2.76±0.16 | 2.46±0.08 | 2.26±0.04 | 3.26±0.25 | 2.86±0.25 | 2.55±0.16 | 3.60 | 0.3 | 16.0 |
| AlphaFlow-PDB | **1.53±0.03** | 1.46±0.01 | 1.41±0.01 | 2.48±0.04 | 2.43±0.01 | 2.40±0.01 | 0.86 | **0.0** | 13.2 |
| AlphaFlow-MD | 1.71±0.08 | 1.51±0.03 | 1.43±0.01 | 2.46±0.09 | 2.32±0.06 | 2.25±0.01 | 1.26 | **0.0** | 26.2 |
| ESMFlow-PDB | 1.59±0.04 | 1.49±0.02 | 1.42±0.01 | 2.49±0.03 | 2.41±0.03 | 2.34±0.01 | 0.74 | **0.0** | 6.0 |
| ESMFlow-MD | 1.68±0.04 | 1.47±0.04 | 1.39±0.03 | 2.44±0.11 | 2.27±0.10 | 2.18±0.02 | 1.17 | **0.0** | 14.3 |
| ConfDiff-Open-MD | 1.64±0.05 | 1.50±0.02 | 1.43±0.02 | 2.50±0.05 | 2.38±0.04 | 2.31±0.02 | 1.37 | 0.2 | **4.6** |
| ConfDiff-Open-ClsFree | 1.66±0.06 | 1.50±0.04 | 1.37±0.02 | 2.56±0.07 | 2.39±0.17 | 2.02±0.10 | 1.77 | 0.5 | 5.5 |
| ConfDiff-ESM-MD | 1.62±0.04 | 1.47±0.02 | 1.40±0.01 | 2.45±0.09 | 2.32±0.05 | 2.25±0.02 | 1.42 | 0.1 | 5.0 |
| ConfDiff-ESM-ClsFree | 1.57±0.04 | 1.45±0.02 | 1.40±0.01 | 2.48±0.04 | 2.40±0.03 | 2.34±0.02 | 1.80 | 0.5 | 7.5 |
| ConfDiff-ESM-Energy | 1.61±0.03 | 1.46±0.02 | 1.42±0.01 | 2.51±0.05 | 2.44±0.03 | 2.40±0.01 | 1.22 | 0.1 | 7.5 |
| ConfDiff-ESM-Force | 1.58±0.04 | **1.43±0.03** | **1.36±0.01** | 2.44±0.06 | 2.35±0.05 | 2.24±0.06 | 1.76 | 0.1 | 8.9 |

### B.2.3 DISTRIBUTION PREDICTION

**[Task Definition]** Distribution prediction challenges models to generate distributions that closely resemble a target distribution, such as the empirical distribution obtained from molecular dynamics (MD) simulations. Unlike previous two tasks, which focus on recovering specific conformations, this task requires models to demonstrate an understanding of "physics and energy" to accurately predict the conformational landscape at the distribution level. This approach further bridges the

Table 14: Performance on the conformation prediction task for the *apo-holo* dataset. *apo/holo*-TM represents the maximum TM-score of the samples relative to the reference *apo/holo* structure. Twenty conformations were sampled for each protein, and the results are reported as mean/median across 91 proteins. The **best** performance is highlighted in bold, and the second-best is underlined. "N/A" indicates not applicable due to model resolution.

| | Accuracy | | | Diversity | Quality | |
| --- | --- | --- | --- | --- | --- | --- |
| | *apo*-TM ↑ | *holo*-TM ↑ | TMens ↑ | Pairwise TM | CA clash % ↓ | PepBond break % ↓ |
| *apo* model | 1.000 | 0.790 | 0.895 | N/A | N/A | N/A |
| EigenFold | 0.831 | 0.864 | 0.847 | 0.907 | 3.6 | N/A |
| MSA-depth256 | 0.845 | 0.889 | 0.867 | 0.978 | 0.2 | 4.6 |
| MSA-depth64 | 0.844 | 0.883 | 0.863 | 0.950 | 0.2 | 5.7 |
| MSA-depth32 | 0.824 | 0.857 | 0.841 | 0.864 | 0.2 | 8.9 |
| Str2Str-ODE ($T_{max} = 0.14$) | 0.762 | 0.778 | 0.770 | 0.954 | 0.2 | 14.0 |
| Str2Str-ODE ($T_{max} = 0.3$) | 0.766 | 0.781 | 0.774 | 0.872 | 0.2 | 14.7 |
| Str2Str-SDE ($T_{max} = 0.14$) | 0.682 | 0.693 | 0.688 | 0.760 | 0.2 | 22.6 |
| Str2Str-SDE ($T_{max} = 0.3$) | 0.680 | 0.689 | 0.684 | 0.639 | 0.2 | 21.1 |
| AlphaFlow-PDB | 0.855 | **0.891** | **0.873** | 0.924 | 0.3 | 6.6 |
| AlphaFlow-MD | **0.857** | 0.863 | 0.860 | 0.894 | 0.2 | 20.8 |
| ESMFlow-PDB | 0.849 | 0.882 | 0.866 | 0.935 | 0.3 | 4.8 |
| ESMFlow-MD | 0.851 | 0.864 | 0.858 | 0.897 | **0.1** | 10.9 |
| ConfDiff-Open-PDB | 0.847 | 0.886 | 0.867 | 0.909 | 0.5 | 5.5 |
| ConfDiff-Open-ClsFree | 0.838 | 0.879 | 0.859 | 0.870 | 0.8 | 5.8 |
| ConfDiff-Open-MD | 0.839 | 0.874 | 0.857 | 0.863 | 0.4 | 6.8 |
| ConfDiff-ESM-PDB | 0.845 | 0.873 | 0.859 | 0.890 | 0.5 | **4.1** |
| ConfDiff-ESM-ClsFree | 0.837 | 0.864 | 0.850 | 0.846 | 0.7 | 4.6 |
| ConfDiff-ESM-MD | 0.836 | 0.862 | 0.849 | 0.846 | 0.3 | **4.1** |

gap between protein conformation prediction models and MD-based methods for studying protein dynamics and thermodynamic properties.

**[Evaluation Metrics]** The **diversity** and **quality** evaluation are the same as in the previous task. We also included the average RMSF (root mean square fluctuation) as an additional metrics for atom-level diversities. To evaluate the **accuracy** of capturing the conformational distribution and dynamics of proteins, we extended the implementation [6] from Jing et al. (2024), with a modification to explicitly align atom orders in `mdtraj` before comparing sample and reference structures. The accuracy are evaluated from three sub-categories:

- **Flexibility:** This metric assesses how accurately the generated samples reflect the protein's flexibility at both the protein and atom levels. It is quantified by the Pearson correlation coefficient ($r$) between the diversity measures, such as Pairwise RMSD for protein or RMSF for atoms, of the model-generated samples and those of the reference MD samples.

- **Distributional accuracy:** This category evaluates the model's accuracy on recovering the target distributions. Wasserstein-2 distances are used to measure the similarity between model-generated and reference distributions. RMWD is the root mean Wasserstein distance between the distributions of aligned coordinates, modeled as multivariate Gaussians. We also evaluate the W2 distance in the PCA projected subspace (PCA W2). Additionally, the cosine similarity between the first principal components from PCA analysis of model-generated and reference conformations serves as another indicator of how well the model captures the correct subspace.

- **Ensemble observables:** Another objective of conformational sampling is to identify certain functionally relevant behaviors (so called *observables*), such as transient residue-residue contacts observed in molecular dynamics. The accuracy on recovering such observables is assessed by comparing those derived from the model generated conformations to reference conformations, using metrics like Jaccard similarity or Spearman correlation.

**[Datasets]** We evaluate performance using the ATLAS dataset (Vander Meersche et al., 2024), a recent database of MD simulation results for diverse proteins. To avoid data leakage for models trained on portions of the ATLAS dataset, we follow Jing et al. (2024) and benchmark on 82 proteins whose PDB entries were deposited after May 1, 2019 and are not part of the training or validation set. ATLAS is a recently published dataset containing triplicated 100 ns MD simulations for 1,390 diverse single-chain proteins. In this work, we use a subset of 82 proteins whose PDB entries were deposited after May 1, 2019, following Jing et al. (2024). "Protein-only" trajectories were

---

[6]`https://github.com/bjing2016/alphaflow/blob/master/scripts/analyze_`
`ensembles.py`

downloaded from the ATLAS database [7] for evaluation. We sample 250 conformations for each protein for evaluation.

**[Model Implementations]** The models are implemented the same as in the multiple-state prediction task. See the previous Section B.2.2 for details.

**[Extended Explanations and Discussion on Model Performance]**

The complete evaluation results for distribution prediction (Atlas) are shown in Table 15. When comparing the effects of structure exploration strategies, classifier guidance used in ConfDiff stands out as the only approach that improves upon the base PDB model. In contrast, strategies such as MSA subsampling and structural perturbation in Str2Str negatively impact most accuracy metrics. However, as discussed in the main text, fine-tuning models on protein conformational data from MD simulations proved to be the most effective strategy, offering significant improvements over the base models. This contrasts with the challenges observed in multiple-state prediction tasks. The likely reason is that capturing conformational distributions requires detailed physical insights, which are difficult to extract solely from PDB structural data.

Table 15: Performance on distribution prediction for the ATLAS test set. A total of 250 conformations were sampled for each protein, and the median values across 82 proteins are reported. The **best** performance is highlighted in bold, and the second-best is underlined. *These metrics require all-atom or backbone predictions; therefore, EigenFold and Str2Str do not have sufficient resolution for evaluation (indicated as "N/A").

| | Diversity | | Flexibility: *Pearson r* on | | | Distributional accuracy | | | |
| | Pairwise RMSD | *RMSF | Pairwise RMSD ↑ | *Global RMSF ↑ | *Per target RMSF ↑ | *RMWD ↓ | MD PCA W2 ↓ | Joint PCA W2 ↓ | PC sim > 0.5 %↑ |
|---|---|---|---|---|---|---|---|---|---|
| MD iid | 2.76 | 1.63 | 0.96 | 0.97 | 0.99 | 0.67 | 0.73 | 0.71 | 93.9 |
| MD 2.5ns | 1.54 | 0.98 | 0.89 | 0.85 | 0.85 | 2.22 | 1.55 | 1.89 | 36.6 |
| EigenFold | 5.96 | N/A | -0.03 | N/A | N/A | N/A | 2.31 | 7.96 | 12.2 |
| MSA-depth256 | 0.83 | 0.53 | 0.25 | 0.34 | 0.59 | 3.60 | 1.79 | 2.91 | 29.3 |
| MSA-depth64 | 2.03 | 1.51 | 0.25 | 0.30 | 0.57 | 4.00 | 1.94 | 3.34 | 18.3 |
| MSA-depth32 | 5.70 | 7.96 | 0.08 | 0.17 | 0.53 | 6.09 | 2.56 | 5.70 | 17.1 |
| Str2Str-ODE ($T_{max} = 0.14$) | 1.66 | N/A | 0.13 | N/A | N/A | N/A | 2.14 | 4.39 | 6.1 |
| Str2Str-ODE ($T_{max} = 0.3$) | 3.15 | N/A | 0.13 | N/A | N/A | N/A | 2.19 | 4.80 | 9.8 |
| Str2Str-SDE ($T_{max} = 0.14$) | 4.74 | N/A | 0.11 | N/A | N/A | N/A | 2.54 | 8.82 | 9.8 |
| Str2Str-SDE ($T_{max} = 0.3$) | 7.54 | N/A | 0.01 | N/A | N/A | N/A | 3.24 | 12.28 | 7.3 |
| AlphaFlow-PDB | 2.58 | 1.20 | 0.27 | 0.46 | 0.81 | 2.97 | 1.61 | 2.61 | 37.8 |
| AlphaFlow-MD | 2.87 | 1.63 | 0.53 | 0.66 | **0.85** | 2.64 | 1.55 | 2.29 | 39.0 |
| ESMFlow-PDB | 2.99 | 1.68 | 0.14 | 0.27 | 0.71 | 4.15 | 1.87 | 3.61 | 28.0 |
| ESMFlow-MD | 3.33 | 2.13 | 0.19 | 0.30 | 0.76 | 3.61 | 1.66 | 3.25 | 25.6 |
| ConfDiff-Open-ClsFree | 3.68 | 2.12 | 0.39 | 0.54 | 0.83 | 2.91 | 1.54 | 2.46 | **46.3** |
| ConfDiff-Open-PDB | 2.89 | 1.43 | 0.38 | 0.51 | 0.82 | 2.96 | 1.59 | 2.46 | 34.1 |
| ConfDiff-Open-MD | 3.43 | 2.21 | **0.59** | **0.67** | **0.85** | 2.75 | **1.41** | **2.27** | 35.4 |
| ConfDiff-ESM-ClsFree | 4.04 | 2.84 | 0.31 | 0.43 | 0.82 | 3.78 | 1.73 | 3.07 | 37.8 |
| ConfDiff-ESM-PDB | 3.42 | 2.06 | 0.29 | 0.40 | 0.80 | 3.62 | 1.68 | 3.13 | 34.1 |
| ConfDiff-ESM-MD | 3.90 | 2.79 | 0.35 | 0.48 | 0.82 | 3.62 | 1.73 | 3.00 | 37.8 |

| | Ensemble observables | | | | Quality | |
| | Weak contacts $J$ ↑ | Transient contacts $J$↑ | *Exposed residue $J$ ↑ | *Exposed MI matrix $\rho$ ↑ | CA clash % ↓ | *PepBond break % ↓ |
|---|---|---|---|---|---|---|
| MD iid | 0.90 | 0.80 | 0.93 | 0.56 | 0.0 | 3.4 |
| MD 2.5ns | 0.62 | 0.45 | 0.64 | 0.25 | 0.0 | 3.4 |
| EigenFold | 0.36 | 0.19 | N/A | N/A | 5.6 | N/A |
| MSA-depth256 | 0.30 | 0.29 | 0.36 | 0.06 | **0.0** | 5.5 |
| MSA-depth64 | 0.38 | 0.28 | 0.40 | 0.16 | **0.0** | 7.6 |
| MSA-depth32 | 0.40 | 0.24 | 0.40 | 0.19 | 0.1 | 11.2 |
| Str2Str-ODE ($T_{max} = 0.14$) | 0.42 | 0.18 | N/A | N/A | **0.0** | 12.1 |
| Str2Str-ODE ($T_{max} = 0.3$) | 0.42 | 0.17 | N/A | N/A | **0.0** | 13.2 |
| Str2Str-SDE ($T_{max} = 0.14$) | 0.40 | 0.13 | N/A | N/A | 0.1 | 21.9 |
| Str2Str-SDE ($T_{max} = 0.3$) | 0.36 | 0.13 | N/A | N/A | 0.2 | 20.2 |
| AlphaFlow-PDB | 0.45 | 0.36 | 0.50 | 0.25 | 0.1 | 6.7 |
| AlphaFlow-MD | 0.62 | **0.41** | **0.69** | **0.35** | **0.0** | 22.2 |
| ESMFlow-PDB | 0.42 | 0.30 | 0.46 | 0.21 | 0.2 | 5.1 |
| ESMFlow-MD | 0.55 | 0.34 | 0.57 | 0.29 | 0.1 | 10.9 |
| ConfDiff-Open-ClsFree | 0.58 | 0.36 | 0.60 | 0.28 | 0.8 | 5.7 |
| ConfDiff-Open-PDB | 0.50 | 0.36 | 0.54 | 0.25 | 0.5 | 5.6 |
| ConfDiff-Open-MD | **0.63** | 0.39 | 0.65 | 0.33 | 0.5 | 6.5 |
| ConfDiff-ESM-ClsFree | 0.57 | 0.34 | 0.59 | 0.23 | 0.9 | 4.3 |
| ConfDiff-ESM-PDB | 0.50 | 0.33 | 0.50 | 0.24 | 0.5 | **4.0** |
| ConfDiff-ESM-MD | 0.61 | 0.36 | 0.61 | 0.31 | 0.4 | 4.3 |

---

[7] https://www.dsimb.inserm.fr/ATLAS/index.html

## C    DETAILED DISCUSSION OF KEY OBSERVATIONS

**Valid evaluation of protein foundation models requires accurate and comprehensive evaluation metrics.** The emergence of folding models like AlphaFold2 and ESMFold offers opportunities for precise assessment of quality, stability, and accuracy in protein generative tasks. However, **certain complex tasks may still lack sufficiently accurate evaluation methods**. For example, within the realm of antibody design, researchers have at times been misled by reconstruction metrics like Amino Acid Recovery (AAR) and Root Mean Square Deviation (RMSD) related to accuracy, resulting in overly optimistic conclusions. In this study, we intend to tackle this challenge by proposing an evaluation strategy integrating reconstitution and physical rationality metrics. Also, we provide a **multifaceted evaluation strategy** to capture various facets of protein structure and function, fostering a more holistic understanding of the performance of foundation models. Furthermore, **metrics alone are insufficient**. In the development of generative models for protein, the primary objective is to accurately fit the distribution of the training data. Our evaluation adopts a more comprehensive strategy that includes measuring the same metrics for the training data (which encompasses native proteins, antibodies, and molecular dynamics conformations in various lengths). This provides a high-resolution gold reference for protein generative targets.

**No single model currently excels across all protein design objectives. The choice of model should be carefully aligned with the intended applications.** In the field of protein foundation models, two primary approaches have emerged: language models and geometric models. Each approach has its strengths and limitations, which are reflected in the performance of ProteinBench. We found language models show good performance in capturing nature evolution distributions, evidenced by their high accuracy in native sequence recovery (inverse-folding) and high quality in scaffolding evolution-conserved motifs. However, language models show limitations in robustness when designing sequences for de novo backbones, and in generating novel sequences for sequence-based protein design. In contrast, structure-based models exhibit greater robustness and tolerance for structural noises in de novo design task, and show greater potential for creating proteins with new folds or functions. These findings underscore the importance of carefully considering specific design objectives when researchers are selecting a model to use.

**While generative models extended from classic folding models have shown an ability to sample protein conformations, challenges remain in both multiple-state prediction and distribution prediction.** Protein conformation prediction is a new but crucial assessment of the multi-modal capabilities and physical understanding of protein foundation models. While strategies proposed in current models may benefit certain tasks, they often provide limited improvement in others. For example, although fine-tuning models using the MD conformation dataset showed promising results on the ATLAS benchmark, little to no improvement was observed in the multi-state prediction of *apo-holo* conformations. Additionally, the common trade-off between diversity and quality in current models underscores the importance of consistent evaluation across the dimensions of accuracy, diversity, and quality in protein conformation prediction tasks.

