# OpenReview forum: "ProteinBench: A Holistic Evaluation of Protein Foundation Models"
_ICLR.cc/2025/Conference — ICLR 2025 Poster_

### Official Review · Reviewer_d7D6 · 2024-10-31

**Soundness:** 2
**Presentation:** 2
**Contribution:** 2
**Rating:** 6
**Confidence:** 4

**Summary:**

The paper introduces a unified protein benchmark to evaluate various methods for different tasks such as protein structure prediction, sequence design, structure design, sequence-structure design, and molecular dynamics. For each task, results are presented for various state-of-the-art methods and a discussion is followed on the strengths/weaknesses of these methods. The evaluation is performed using multiple metrics commonly used for these tasks, and, that sometimes incorporate different objectives in protein design. The benchmark is planned to be shared in an open-source manner for the community to compare methods for the presented tasks.

**Strengths:**

1.	A benchmark with multiple metrics is proposed for protein-related tasks. The metrics incorporate different objectives for different problems encountered in protein design and molecular dynamics tasks.
2.	The paper tackles an important and complex problem in the protein community which is the creation of benchmarks and datasets for training AI methods.
3.	The benchmark incorporates recent challenges like evaluating sequence-structure compatibility in recent co-design methods.

**Weaknesses:**

1.	The evaluation of methods for protein-related tasks is challenging and, usually, the decision of dataset curation and splits, metrics used for evaluation, and other small details are very important so the results can be trusted by the community. The reviewer thinks that the manuscript could have more information on all these details, instead of presenting results and discussing the performance of the methods. Some of these details are presented in the Appendix while others are missing.
2.	Some metrics used for the benchmark, even though used before by previous references, might need more discussion as they can be misleading for checking the quality of designs, e.g. scRMSD in antibody design.

**Questions:**

Comments:

1.	My main concern is related to how the paper is structured and the information it contains. From Page 3 the task results are presented with a small definition and comparison between methods. I see that the authors presented the creation of unified datasets as a current limitation and future work by the community, but the reviewer thinks that the manuscript should contain more information about the thought process about creating the benchmark, with the thought process about the metrics, the impact of methods using different datasets, tasks in which this is critical, etc. These are crucial for the community to adopt the benchmark and trust the results that are being presented and compared. In its current form, it is hard to understand these intrinsic details that are important for protein-related tasks.
2.	The authors define the methods as “protein foundation models” and add their own explanation of how this definition is being used. From the reviewer's understanding, usually, foundation models are defined for methods that can be applied, e.g. using their latent space, for many different tasks.  Any additional reasoning for using this new definition of protein foundation models?
3.	Motif Scaffolding: For the Motif Scaffolding task, which evaluation metrics are being used? The reviewer is confused if RMSD metrics are being used or also designability metrics are being used.
4.	Antibody Design: Antibody metrics are usually very challenging to trust. As a benchmark, some of the metrics such as the scRMSD from structure prediction networks like IgFold can be misleading, as we are more interested in antigen-antibody complex structures. I understand current structure prediction networks accuracy for antibodies is limited, but it would be interesting to discuss and choose reliable metrics, even at a reduced number. As the evaluation metrics evolved, they could be added to the benchmark.

Minor Comments:

1.	Line 81: “we aims”
2.	Lines 193-195: “We have noticed…” and Lines 214-215: “We will soon…”: These sentences can be re-written to mention or list just current state-of-the-art methods that are currently not evaluated.

---

> ### Author Response · Authors · 2024-11-21
> **Rebuttal by Authors**
>
> We greatly appreciate the reviewer's constructive feedback regarding the paper structure and implementation details of the benchmark. In response, we have implemented two major improvements: First, we have thoroughly restructured the manuscript and appendix using a task-centric approach, ensuring clearer organization and logical flow of information. Second, we enhanced our analytical depth by providing comprehensive discussions and deeper insights for each protein design task. We believe these revisions substantially strengthen the manuscript and welcome your assessment of these changes.
>
> **[Q1]** The evaluation of methods for protein-related tasks is challenging and, usually, the decision of dataset curation and splits, metrics used for evaluation, and other small details are very important so the results can be trusted by the community. The reviewer thinks that the manuscript could have more information on all these details, instead of presenting results and discussing the performance of the methods. Some of these details are presented in the Appendix while others are missing.
>
> **[A1]** We thank the reviewer's thoughtful comment regarding evaluation transparency. In our revised manuscript, we have expanded the implementation details for each method to ensure reproducibility and trustworthiness of our results. We have ensured that all critical details regarding dataset curation, data splits, evaluation metrics, and implementation specifications are clearly presented. These details are now comprehensively documented in the appendix Section B lighted in blue.

---

> ### Author Response · Authors · 2024-11-21
> **Rebuttal by Authors**
>
> **[Q2]** Some metrics used for the benchmark, even though used before by previous references, might need more discussion as they can be misleading for checking the quality of designs, e.g. Antibody Design: Antibody metrics are usually very challenging to trust. As a benchmark, some of the metrics such as the scRMSD from structure prediction networks like IgFold can be misleading, as we are more interested in antigen-antibody complex structures. I understand current structure prediction networks accuracy for antibodies is limited, but it would be interesting to discuss and choose reliable metrics, even at a reduced number. As the evaluation metrics evolved, they could be added to the benchmark.
>
> **[A2]** Thanks for your suggestions.
>
> There are some misunderstandings here. We totally agree with the point that what should be focused on is the complex structure instead of the isolated antibody structure and this is why we obtain the reference structure used for scRMSD calculation in a two-stage way.
>
> In the first stage, IgFold is utilized in the first stage to get an initial antibody CDR structure (the predicted isolated antibody structure). The initial CDR structure then undergoes a structure optimization for lower energy with the condition of the antigen-antibody complex (the simulated antigen-bound antibody structure). Thus, the first stage is antigen irrelevant while the second stage is antigen dependent.
>
> We have also tried to use AF2-multimer to obtain the reference structure but finally gave up because of the huge time consumption in the MSA building for antibody sequence.
>
> The two-stage strategy is inspired by the slight interface structural changes in antigen-antibody binding [1] for lower energy. RMSD on reference antibodies also proves our strategy, which is reduced from 1.9* to 1.7* after the second stage. We will also keep updating our benchmarks using more reliable methods, like we will switch the calculation of scRMSD based on AF3 once it is completely open access.
>
> Reference:
>
> [1] Guest, Johnathan D., Thom Vreven, Jing Zhou, Iain Moal, Jeliazko R. Jeliazkov, Jeffrey J. Gray, Zhiping Weng, and Brian G. Pierce. "An expanded benchmark for antibody-antigen docking and affinity prediction reveals insights into antibody recognition determinants." Structure 29, no. 6 (2021): 606-621.

---

> ### Author Response · Authors · 2024-11-21
> **Rebuttal by Authors**
>
> **[Q3]** My main concern is related to how the paper is structured and the information it contains. From Page 3 the task results are presented with a small definition and comparison between methods. I see that the authors presented the creation of unified datasets as a current limitation and future work by the community, but the reviewer thinks that the manuscript should contain more information about the thought process about creating the benchmark, with the thought process about the metrics, the impact of methods using different datasets, tasks in which this is critical, etc. These are crucial for the community to adopt the benchmark and trust the results that are being presented and compared. In its current form, it is hard to understand these intrinsic details that are important for protein-related tasks.
>
> **[A3]**  In our revised version, we provide a manuscript structure where the main manuscript presents key evaluation results, enabling readers to quickly grasp the overall landscape of tasks and performance of protein foundation models. For researchers interested in detailed task motivations, thoughts on metric selections, dataset considerations, implementation information, and result analysis, we have placed the detailed information in the appendix.
>
> Specifically, we have reorganized the appendix (Section B) using a task-centric approach. For each task, we now provide comprehensive details including:
> 1. Task definitions
> 2. Metric justification and descriptions (including the thought process of metrics)
> 4. Dataset specifications
> 5. Supplementary results
> 6. Potential discussions to provide more insights
>
> To facilitate navigation, we added direct links in each task section that connect to their corresponding detailed explanations in the appendix. The reorganized information is now highlighted in blue for improved visibility. Additionally, we will also release a task-centered arxiv paper. This format allows researchers to access task-specific information efficiently.

---

> ### Author Response · Authors · 2024-11-21
> **Rebuttal by Authors**
>
> **[Q4]** The authors define the methods as “protein foundation models” and add their own explanation of how this definition is being used. From the reviewer's understanding, usually, foundation models are defined for methods that can be applied, e.g. using their latent space, for many different tasks. Any additional reasoning for using this new definition of protein foundation models?
>
> **[A4]** Foundation models are traditionally defined as models capable of performing multiple diverse tasks. In our study, we offer a broad definition of "protein foundation models," and our reasoning is grounded in two key observations from the protein science domain:
>
> 1. protein-related tasks are inherently diverse, flexible, and complex. This is evident in cases like inverse folding, where a single task can serve multiple distinct applications, each with different metrics priorities. Similarly, in protein sequence-structure co-design, methods developed for this task demonstrate remarkable versatility. They can generate sequences, predict structures, or simultaneously accomplish both objectives. **This intrinsic task flexibility means that even methods initially designed for specific applications often demonstrate broader utility across multiple tasks.**
>
> 2. Our benchmark includes established foundation models like AlphaFold, which has proven capabilities in both protein structure prediction and conformation prediction. While many early methods were indeed specialized and achieved exceptional performance in specific tasks, their practical applications often extend beyond their original scope. **To create a comprehensive and practical benchmark, we have chosen to include all relevant methods regardless of their original design intent.** This inclusive approach better reflects the current state and practical utility of protein modeling methods.

---

> ### Author Response · Authors · 2024-11-21
> **Rebuttal by Authors**
>
> **[Q5]** Motif Scaffolding: For the Motif Scaffolding task, which evaluation metrics are being used? The reviewer is confused if RMSD metrics are being used or also designability metrics are being used.
>
> **[A5]** Thanks for the question. We measure motif-scaffolding in terms of both motif accuracy and overall designability. For motif accuracy, we calculate RMSD between the input motif structure and the corresponding region of the designed protein to assess whether the motif structure is preserved (motifRMSD < 1.0). As for the overall designability, we use scTM score > 0.8 as being designable. We have accordingly elaborated on motif-scalffolding evaluation in the appendix.

---

> ### Author Response · Authors · 2024-11-21
> **Rebuttal by Authors**
>
> **[Q6]** Minor Comments:
> 1. Line 81: “we aims”
> 2. Lines 193-195: “We have noticed…” and Lines 214-215: “We will soon…”: These sentences can be re-written to mention or list just current state-of-the-art methods that are currently not evaluated.
> We thank the reviewer for bringing this to our attention. In the revised manuscript, we have corrected the typo errors in line81, and polished statements in lines193-195 and lines 214-215.
>
> **[A6]** In the revised manuscript, we have expanded our evaluation to include Proteous's performance across multiple protein lengths (100, 200, 300, and 500 residues). Our analysis reveals that Proteous demonstrates superior design quality for long-chain backbone design (500 residues), achieving an scTM score of 0.90 compared to RFdiffusion's 0.79. However, we observed a significant decline in structural diversity for Proteous when designing longer chains:
>
> - At 300 residues: Proteous diversity score 0.34 vs. RFdiffusion 0.65
> - At 500 residues: Proteous diversity score 0.34 vs. RFdiffusion 0.89
>
> *Note: We are still doing the evaluation and expect to finish all the result in next few days.*
>
> Case analysis revealed that Proteous tends to generate structures limited to three categories, predominantly characterized by helical tandem repeats, confirming our diversity metric findings. Detailed discussion of these results is provided in Appendix B.1.2-PROTEIN BACKBONE DESIGN-[Extended Explanations and Discussion on Model Performance].
>
> We have also incorporated performance evaluation results for CarbonNovo in the structure-sequence co-design section. Given the rapid pace of new publications in this field, maintaining completely up-to-date evaluations presents a challenge. Nevertheless, we are committed to continuous updates of our benchmark to support advancement in the field.

---

> > ### Comment · Reviewer_d7D6 · 2024-11-23
> > **Rebuttal Feedback**
> >
> > The reviewer thanks the authors for addressing part of my concerns and comments.
> >
> > I have increased my score.
> >
> > I still do think that the most important content of the manuscript is in Appendix B. I understand the page limitations of the manuscript and the possibility of uploading an Arxiv version with the re-structured content, however, I feel that as a part of the protein community and as a user of the benchmark for both comparing novel proposed algorithms and choosing the best algorithm for a specific application, it is very important that the most important details and reasoning for each task regarding evaluation/fairness/datasets are in the manuscript. The benchmark results will change when new algorithms are proposed, but, in the reviewer's opinion, the benchmark will be strong and pass the test of time if the metrics and dataset preprocessing remain the most stable over time, especially in the protein domain, where many works are currently being developed for benchmarks/datasets to address limitations in the field, e.g. https://www.biorxiv.org/content/10.1101/2024.07.17.603980v2 .

---

> > > ### Comment · Reviewer_bamz · 2024-11-24
> > > **Agreed. The paper (especially the main text) needs more and better motivation of it's choices**
> > >
> > > I agree with this reviewer, who made similar points to those I made. To stand the test of time this benchmark needs to motivate and explain the major tests it includes, rather than focus so much energy on the results of current models, which will quickly become outdated as a new better models emerge. The authors seem to have understood the point, but only made most of the changes in the appendix. Thus this reviewer and I agree that most of the important and best parts of this paper are buried in the appendix, rather than being the main text where they should be. For this reason I am leaving my store where it was. This reviewer's comments actually made me think my original score probably was too high, as the paper really isn't doing a very good job of explaining the motivation for each of the challenges in this benchmark.

---

> > > ### Author Response · Authors · 2024-11-24
> > > **Rebuttal by Authors**
> > >
> > > First of all, many thanks to the reviewer for increasing the score. We also understand the reviewer's concern about the structure of the manuscript. We rationalize the organization of the paper for the following reasons.
> > >
> > > 1. As highlighted in Appendix A, one of the most important contributions of ProteinBench is the **first comprehensive benchmark for protein foundation models**, differentiating this work from the existing benchmarks targeting specific tasks. We have so much content that should be presented in the limited pages of the manuscript. In our revised version, we aim to present the **general comprehensive landscape** for all 8 different protein tasks. The main manuscript is organized to deliver the most important general information for all tasks by providing high-level design logic summarized in Table 1 and comparative studies of existing state-of-the-art models' performance. **To improve the reader experience, we provide the links in Table 1 in the updated version to quickly navigate to valuable details in the appendix. The general landscape allows the users of the benchmark, who are familiar with some of the tasks, to quickly find the methods they would like to use and compare with.**
> > >
> > > 2. We acknowledge the importance of providing detailed information about datasets and evaluation metrics and upholding such standards for testing future models. **At the same time, we recognize that comparative studies among models are a valuable contribution at the current stage of the field and should be included in the main text. These insights could be essential for guiding the current and future development of models. This is especially critical in emerging areas like protein conformation prediction, where no standardized comparisons of existing models have been conducted to evaluate their performance across different tasks.**
> > >
> > > 3. Due to the **page limitations**, we have to put all the details in the appendix. **We understand the details are important to the users, especially the users who are not protein experts to understand the tasks. To help those users to grasp each task from scratch, we provided as much detailed information as we could.** If we put the information for all protein tasks in the main part, users may get lost in the details.
> > >
> > > 4. We thank the reviewer for providing a new protein interaction benchmark PINDER for our reference (https://www.biorxiv.org/content/10.1101/2024.07.17.603980v2) and we recruited this paper in the revised version. This benchmark provides a new dataset specifically targeting the evaluation of complex structure prediction. We want to emphasize the contribution of our benchmark is comprehensively assessing protein foundation models and their ability to understand multiple modalities.

---

> > > > ### Comment · Reviewer_d7D6 · 2024-11-25
> > > > **Rebuttal Feedback 2**
> > > >
> > > > Thanks for providing the rationale behind the organization of the manuscript.
> > > >
> > > > I understand and agree with the contributions mentioned by the authors that the manuscript provides an important benchmark for various protein-related tasks.
> > > >
> > > > I still keep my opinion that for protein-related benchmarks, especially for tasks with no standardized comparisons, having clear discussion and reasoning of datasets and metrics is more critical than providing only comparative studies. For example, if the datasets contain data leakage or if the metrics do not have any correlation to wet lab experiments, these comparative studies might mislead readers.
> > > >
> > > > I am open to discussing with other reviewers with diverging opinions regarding the paper structure. At this moment I decide to keep my score.

---

> ### Author Response · Authors · 2024-11-25
> **Rebuttal by Authors 2**
>
> Thanks for further discussion.  To address the reviewer's concerns, we provide more facts to rationalize our detailed implementation of datasets and metrics and paper organization.
>
> 1. We acknowledge the importance of providing discussion and reasoning of datasets and metrics, and **we have provided all the detailed information in Appendix B.**
>
> 2. **We selected datasets and metrics that have been tested in the field of each task to ensure the quality of our benchmark.** Many of the datasets and metrics we used are standardized datasets and standardized metrics have been widely used in previous studies. For example, CASP and CAMEO datasets are well-recognized datasets widely used in the protein structure prediction field. The released-date-based data split is well-accepted in the field to avoid data leakage. Self-consistency TMscore/RMSD are widely used for protein design. Ensemble TMscore/RMSD for accuracy in the multiple-sate conformation or Pairwise RMSD/RMSF. We carefully followed the standardized processing procedures in our implementation. All the details are provided, and reference papers are cited in the appendix.
> **However, although standardized datasets and metrics have been introduced, the field still lacks a comprehensive multi-metric evaluation approach that assesses performance across different datasets for protein foundation models. Thus, we focused on the comparative study in the main manuscript.**
>
> 3. Another fact is **many of the protein foundation models evaluated in this study are generative models that do not rely on the usage of test datasets for evaluation.** These models are included in three protein design tasks (backbone design, sequence design, and co-design).
>
> 4. For the antibody design task, new metrics are specifically introduced in detail in the appendix. Also, to avoid the risk of data leakage, we carefully trained and tested all the models using a unified dataset with implementation details carefully introduced.
>
> 5. **If the reviewer has specific suggestions regarding the reorganization of the paper, please let us know, and we will consider adopting them.**
>
> We hope these facts can release the reviewer's concerns about the datasets and metrics.

---

> > ### Comment · Reviewer_d7D6 · 2024-11-26
> > **Rebuttal Feedback 2**
> >
> > Thanks for addressing my comments and discussing the paper's content and organization.
> >
> > Regarding the organization, my specific suggestion would be to not focus too much on describing individual results, but also add important information currently in the appendix. For example, trying to answer questions from readers like: (i) Which datasets and splits should I use if I want to add my algorithm to the benchmark or compare with the results for other methods?; (ii) Which metrics and how should I use them for these tasks?; (iii) Training Strategy influences Robustness; etc.
> >
> > As these comments can be biased by a personal preference in the organization/presentation of the manuscript and I recognize the efforts by the authors to improve the completeness of Appendix B, **I am changing my score to 5.5 (in the acceptance threshold)**. In summary, I recognize the strengths of this paper and its importance for the community, even though I think the presentation could be improved as a manuscript.
> >
> > Minor comments:
> > 1. There are still a few typos and problems with symbols to be corrected after the additions during the Discussion process.
> > 2. The sampling temperature was fixed at 0.1 for all inverse folding methods, but the optimal value can vary for different methods.

---

> > > ### Author Response · Authors · 2024-11-27
> > > **Further response to reviewer d7D6**
> > >
> > > Thank you for your specific suggestions and for raising the rating to 5.5. We now fully understand your feedback regarding the reorganization of the paper. We agree that the task specifications (e.g., datasets, splits, metrics, etc.) in the appendix are essential for readers interested in these low-level design details, while those with extensive experience in protein-related tasks would prefer a greater emphasis on the comparative performance analysis of standardized benchmarks to guide model development. As you noted, this preference largely depends on the background and personal preferences of the readers.
> > >
> > > Following the reviewer’s suggestions, we attempted to move the task specifications into the main text. However, to adhere to the 10-page limit, we had to relocate 2 of the 8 tasks to the appendix, which we believe compromises the overall comprehensiveness of the benchmark. Therefore, we decided to retain the current organization of the paper. This decision should not be interpreted as disregarding your suggestion; it is solely a compromise due to the page limit. Our intent was to present a holistic view of our protein benchmark within the main paper.
> > >
> > > With that said, to better serve the needs of different readers, we will prepare an extended version of the paper that exceeds the 10-page main text limit. This version will strictly follow your suggestion to integrate task definitions, technical details, and performance evaluations into the main text. We will release this extended version on arXiv and our GitHub repository as a supplement to the ICLR camera-ready version.
> > >
> > > Regarding your minor comments:
> > >
> > > 1. The typo errors have been corrected in this revision, and we will conduct a final thorough review for the camera-ready version.
> > >
> > > 2. Regarding sampling temperature, we acknowledge the diversity-quality trade-off and chose 0.1 as a default value, while recognizing that optimal values may vary depending on the specific inverse folding method and the intended design goals (e.g., prioritizing diversity versus quality). We have added a short comment on this point in the revision.
> > >
> > > We sincerely appreciate the reviewer’s detailed suggestions and feedback. We kindly request the reviewer to consider raising the score to a solid 6 (as 5.5 is not a valid rating in ICLR), which would provide us the opportunity to serve this benchmark to the broader protein research community.

---

> > > > ### Comment · Reviewer_d7D6 · 2024-11-27
> > > > **Rebuttal Feedback 3**
> > > >
> > > > Thank you for the attempt to address my comments on the paper structure.
> > > >
> > > > 1. I tend to disagree with the sentence by the authors that readers "with extensive experience in protein-related tasks would prefer a greater emphasis on the comparative performance analysis of standardized benchmarks". As an interdisciplinary field with readers comprising computer scientists, computational biologists, and biologists it is important to have a benchmark that is trusted by biologists while guiding model development by computer scientists and computational biologists.
> > > >
> > > > 2. The authors created the benchmark comprising both protein design and protein conformation prediction tasks. I understand that having more tasks is an effort by the authors to have a unified benchmark. But it also made me think that if the focus was only on protein design tasks the delivery of the contents of the manuscript would probably be stronger (addressing my comments and comments from reviewer bamz without the need for an extended version on Arxiv).
> > > >
> > > > 3. Regarding the sampling temperature, thanks for adding a comment on this point. I think this is critical because methods evaluated in the benchmark might think it is unfair to use a temperature value that is optimized for a specific method. In the future, it would be better to use the optimal values from each reference paper or evaluate these models for different temperature values.
> > > >
> > > > I will increase my score to 6 however I will keep my concerns regarding the presentation of this manuscript as I think that a robust presentation of this benchmark can have a big impact on the protein research community, especially in the protein design field.

---

> > > > > ### Author Response · Authors · 2024-11-28
> > > > > **Further response to reviewer d7D6**
> > > > >
> > > > > We sincerely thank the reviewer for raising the score. Your perceptive remarks and thoughtful feedback have been invaluable, not only in improving the quality of our work but also in guiding the future optimization of the protein benchmark.
> > > > >
> > > > > As previously discussed, reader preferences are diverse. Protein conformation is an emerging field of research that currently lacks comprehensive comparative analysis. Our work holds the potential to be beneficial to researchers engaged in this area at the present stage.
> > > > >
> > > > > To better meet the needs of diverse readers, we are committed to preparing an extended version of the paper, which will fully incorporate your suggestions, integrating task definitions, technical details, and performance evaluations directly into the main text for greater clarity and accessibility.
> > > > >
> > > > > We are truly appreciative of the reviewer's comment regarding the sampling temperature. Future evaluations and in-depth analyses of the optimal sampling temperatures for different methods will be conducted.

---

### Official Review · Reviewer_hosV · 2024-11-01

**Soundness:** 3
**Presentation:** 2
**Contribution:** 2
**Rating:** 6
**Confidence:** 4

**Summary:**

The paper  introduces ProteinBench, an evaluation framework aimed at standardizing and broadening the assessment of protein foundation models. These models have gained prominence due to advancements in protein prediction and design, covering tasks from structural prediction to conformational dynamics. ProteinBench aims to address gaps in current evaluation practices by introducing a comprehensive, multi-dimensional approach that evaluates models based on quality, novelty, diversity, and robustness. The authors aim for ProteinBench to become a continually updated benchmark to guide research and collaboration in protein modeling.

**Strengths:**

Novel Evaluation Framework: The paper proposes a well-structured framework that standardizes evaluation for protein foundation models, addressing a significant need in the field. By evaluating on multiple fronts—quality, novelty, diversity, and robustness—ProteinBench gives a well-rounded assessment of model performance.

Task Diversity and Practical Relevance: ProteinBench is inclusive of various protein modeling tasks, including antibody design and multi-state prediction, which are highly relevant to real-world applications in pharmaceuticals and bioengineering.

User-centered Analysis: The framework is flexible, accommodating different user needs (e.g., evolutionary fidelity vs. novelty), which makes the tool versatile for diverse research goals. This feature improves the applicability of model results to specific scientific or engineering contexts.

**Weaknesses:**

Lack of Standardized Training Data: Differences in training datasets among models hinder direct comparison. Standardizing datasets would improve the ability to compare model architectures and may be essential for achieving fairer assessments within ProteinBench.

**Questions:**

1. I am not very familiar with AI for Protein. Could you provide with the reason why you separate whole protein tasks to these 8 parts?

2. Could you give some additional metric about tasks? While ProteinBench is a strong foundation, additional tasks and metrics (especially regarding dynamics and multi-modal integrations) could improve its scope, making it more universally applicable in protein science.

3. Could you plan to implement controls to account for differences in training data?

---

> ### Author Response · Authors · 2024-11-21
> **Rebuttal by Authors**
>
> We greatly appreciate your constructive feedback regarding the consistency of the training dataset and the design rationale of the benchmark. In response, we have provided additional information for your major concerns: First, we recognize the importance of controlling for training data differences and plan to implement more rigorous controls to account for differences in training data. Second, we restructured the manuscript and provided additional discussion about the design rationale of the benchmark.
>
> **[Q1]** Lack of Standardized Training Data: Differences in training datasets among models hinder direct comparison. Standardizing datasets would improve the ability to compare model architectures and may be essential for achieving fairer assessments within ProteinBench. Could you plan to implement controls to account for differences in training data?
>
> **[A1]** We thank the reviewer for bringing this to our attention.
>
> 1. **Our current benchmarking approach focuses on evaluating existing methods and models at the model layer rather than the technique layer**, where training data is considered an integral part of each method's strategy. This approach aligns with other established foundation model benchmarks that standardize model evaluation rather than isolating technical components. We believe this better serves users by providing insights into real-world model performance.
>
> 2. In our manuscript, **all models for the antibody design task were retrained using the same dataset**. This consistency enables a direct comparison of their underlying technical approaches. However, for the other tasks, the training datasets varied, which may affect the comparability of the techniques.
>
> 3. We recognize the importance of controlling for training data differences. We envision ProteinBench as an evolving benchmark, and in future iterations, **we plan to implement more rigorous controls to account for these differences in training data.** This will help provide deeper insights into the impact of training data on model performance.

---

> ### Author Response · Authors · 2024-11-21
> **Rebuttal by Authors**
>
> **[Q2]** I am not very familiar with AI for Protein. Could you provide with the reason why you separate whole protein tasks to these 8 parts?
>
> **[A2]** Many thanks for the question.
>
> **1. Why do we focus on protein design and conformation prediction?**
>
> Protein three-dimensional structure prediction has become a pivotal area of research, leading to established benchmarks such as CASP and CAMEO, along with significant methodological advancements, including the AlphaFold series, RosettaFold, ESMFold, OmegaFold, and others. While structure prediction addresses the challenge of determining a protein's structure from known sequences, protein design represents the inverse problem: predicting sequences that will fold into specific structures or fulfill designated functions.
>
> Despite the increasing interest in protein design, the field currently lacks a comprehensive benchmark, which has limited community progress. Existing benchmarks, as documented in Appendix Table 1, primarily focus on specialized tasks. A similar gap exists in the realm of conformational dynamics research. To address these critical issues, we present the first comprehensive benchmark that emphasizes two fundamental tasks: protein design and conformation prediction.
>
> **2. Why separate protein design into five parts?**
>
> The scientific scope allows us to naturally divide protein design into five key categories following the sequence-structure-function hierarchy:
> - Sequence design: Optimizing amino acid sequences for stable folding.
> - Backbone design: Engineering the overall architecture of the protein.
> - Sequence-structure co-design: A challenging task to simultaneously generate sequence and structure
> - Motif scaffolding (Function design): Incorporating functional motifs into stable scaffolds
> - Antibody design (Function design): Specialized design of antibody structure and sequences for antigen binding. An important application in therapeutic antibody development.
>
> **3. Why separate protein conformation prediction into three parts?**
>
> Again, the scientific scope allows us to naturally divide conformation prediction into three parts based on their biological reality and complexity:
>
> **- Single conformation prediction:** Proteins existing in different conformations. Predict the single dominant state is to identify the lowest energy conformation.
>
> **- Multiple conformation prediction:** A more complicated task to predict discrete conformational states.
>
> **- Conformational Distribution Prediction:** This task is more challenging focusing on the prediction of probability distribution of conformations.
>
> **To benefit non-experts, we have added the description of the rationale in our revised manuscript in Appendix Section A1, as follows:**
>
> 'The field of protein three-dimensional structure prediction has witnessed remarkable progress, exemplified by established benchmarks like CASP and CAMEO, and breakthrough methodologies including AlphaFold series, RosettaFold, ESMFold, and OmegaFold. While structure prediction focuses on determining protein structures from known sequences, protein design addresses the inverse challenge: creating sequences that will fold into desired structures or achieve specific functions. Despite growing interest in protein design, the field has been hampered by the absence of a comprehensive benchmark, with existing evaluations primarily targeting specialized tasks, as documented in Appendix Table 1. A similar limitation exists in conformational dynamics research. Our work addresses these gaps by introducing the first comprehensive benchmark focusing on protein design and conformation prediction.
>
> In our benchmark, protein design is categorized into five distinct areas, following the natural sequence-structure-function hierarchy. This begins with sequence design, focusing on optimizing amino acid sequences for stable folding, and progresses to backbone design, which involves engineering the overall protein architecture. The more complex sequence-structure co-design task requires simultaneous optimization of both sequence and structure. At the functional level, motif scaffolding involves incorporating functional motifs into stable scaffolds, while antibody design represents a specialized application focusing on engineering antibody structures and sequences for antigen binding, particularly crucial for therapeutic development.
>
> The conformation prediction component is similarly structured into three distinct categories, reflecting increasing levels of complexity in protein dynamics. Single conformation prediction focuses on identifying the lowest energy state among possible conformations. Multiple conformation prediction addresses the more complicated challenge of predicting discrete conformational states. The most sophisticated category, conformational distribution prediction, tackles the complex task of predicting probability distributions of conformations, essential for understanding proteins with dynamic structural ensembles.'

---

> ### Author Response · Authors · 2024-11-21
> **Rebuttal by Authors**
>
> **[Q3]** Could you give some additional metric about tasks? While ProteinBench is a strong foundation, additional tasks and metrics (especially regarding dynamics and multi-modal integrations) could improve its scope, making it more universally applicable in protein science.
>
> **[A3]** We thank the reviewer for highlighting the importance of task breadth and evaluation metrics. In curating the benchmark, we thoroughly reviewed the current literature in each domain to ensure comprehensive coverage of tasks and metrics using publicly accessible datasets. **We collected the most representative datasets and tasks to the best of our knowledge.** However, in emerging areas like protein conformational dynamics, such datasets remain limited. That said, we anticipate that future developments in this area will provide more datasets, allowing us to evaluate models with additional tasks and perspectives. We remain committed to the continuous development and maintenance of ProteinBench, incorporating new integrations, updates, and revisions of tasks and metrics.
> **As the whole field keeps evolving. We hope proteinBench is an evolving benchmark, and we will include more tasks and metrics in the future.**

---

> > ### Comment · Reviewer_hosV · 2024-11-24
> > **Official comment by Reviewer hosV**
> >
> > I believe you have addressed my concerns on novelty, training stage, protein tasks and metrics, so I have decided to increase my rating.

---

### Official Review · Reviewer_bamz · 2024-11-02

**Soundness:** 3
**Presentation:** 2
**Contribution:** 3
**Rating:** 6
**Confidence:** 2

**Summary:**

I am not sure how to review this manuscript. I have been in the field for over 20 years, so I am an experienced ML/Deep learning researcher, but I do not know much about deep learning for protein design/generation beyond having read blog posts about AlphaFold, ESM, and similar advances. I do know a fair bit about biology (probably more than most ML researchers), but I am still far from an expert in this area. I thus have some high-level opinions about the work, but cannot evaluate the vast majority of the content. I would like my high-level thoughts to be considered, but I cannot evaluate any of the details in the paper, so I have to defer to other expert reviewers to do that. Here are examples of the questions I cannot evaluate: Are the challenges in the benchmark the right ones? Are they comprehensive enough? Are the models evaluated the right ones? Are major ones missing? Were the evals done fairly and correctly? All of that I’ll have to leave for subject matter experts.

That said, here are my high-level thoughts:

The field benefits from benchmarks, and we should reward people who take the time to make them. I thus support publication because from what I can tell, this is a new, needed, helpful benchmark. However, I would not know if other similar benchmarks already exist.
It is also helpful to have a set of leading models tested on such benchmarks. Assuming the set of models is a good one and the evals are well done, that is another contribution worthy of sharing with the community via a publication. That said, I do not know how novel such a comparison is.
The paper does a poor job of explaining almost anything in a way that a non-domain expert could understand. The worst example of this is the challenges in the benchmark. They are extremely superficially described, with a reference to the Appendix presumably doing all the work of explaining what these challenges really are, how they are evaluated, why they matter, why they were chosen, etc. I think more of that belongs in the paper. I’d rather see a paper arguing for and introducing a benchmark do MUCH more of that motivation and explanation than a slog through the performance on the benchmark of a bunch of models. After all, why do we care how these models perform on problem x before we know what problem x is and why it matters? That said, perhaps domain experts already know these challenges so well that the current level of text is sufficient? I am not in a position to judge, but I think the paper would likely be higher-impact and better if it was more readable by non-insiders. I was hoping to learn a LOT about this area by reading this manuscript, and instead, sadly, I learned basically nothing (except there is a new benchmark). The reason I am still voting for publication despite that is I imagine this paper is quite valuable for insiders, but great papers do better at offering *something* to non-insiders.

Minor:
You say ProteinBench provides a holistic view. That is too strong. It is virtually impossible to provide a holistic view of such a complex topic. Please switch such language to “more holistic” or similar.
You say benchmarks are crucial or critical for progress. This is a strong claim. I think they are helpful, but progress can be made without them, and they can actually hurt progress too (see Why Greatness Cannot Be Planned), so I recommend a more nuanced statement.
Line 181: You say this finding has significant implications for the field, but it is drawn from 2 data points only! Please properly soften the claims.
Line 192: “We have noticed that….” That’s a weird way to describe your own work/choices. You sound surprised you didn’t include more methods? I suggest finding clearer, less confusing language.

Note: I am rating this a 6 and not an 8 simply because I do not know enough to evaluate the vast majority of the work. If the other reviewers think it is good, then this 6 should support publication. But I do not want to set a super high score that would override experts with doubts. If all experts agree the technical work and novelty are solid, I'd be happy with an 8.

**Strengths:**

See main review.

**Weaknesses:**

See main review.

**Questions:**

See main review.

---

> ### Author Response · Authors · 2024-11-21
> **Rebuttal by Authors**
>
> We greatly appreciate your constructive feedback regarding the manuscript's organization, lack of clarity for non-experts, and novelty of the benchmark. In response, we have implemented two major improvements: First, we have thoroughly restructured the manuscript and appendix using a task-centric approach, ensuring clearer organization and logical flow of information. Second, we enhanced our analytical depth by providing comprehensive discussions and deeper insights for each protein design task. We believe these revisions substantially strengthen the manuscript and welcome your assessment of these changes.
>
> **[Q1]** The field benefits from benchmarks, and we should reward people who take the time to make them. I thus support publication because from what I can tell, this is a new, needed, helpful benchmark. However, I would not know if other similar benchmarks already exist. It is also helpful to have a set of leading models tested on such benchmarks. Assuming the set of models is a good one and the evals are well done, that is another contribution worthy of sharing with the community via a publication. That said, I do not know how novel such a comparison is.
>
> **[A1]** We thank the reviewer for raising this important question. In the revised manuscript, we outline the rationale behind our benchmark and provide an overview of existing benchmarks in Appendix A. Existing benchmarks for protein design and conformation prediction are summarized in Table 11. Our analysis indicates that current benchmarks primarily focus on specific tasks, underscoring the urgent need for a comprehensive benchmark that addresses a wider range of protein tasks. In this study, we present **the first extensive benchmark for protein foundation models, covering a broad spectrum of protein tasks**.

---

> ### Author Response · Authors · 2024-11-21
> **Rebuttal by Authors**
>
> **[Q2]** The paper does a poor job of explaining almost anything in a way that a non-domain expert could understand. The worst example of this is the challenges in the benchmark. They are extremely superficially described, with a reference to the Appendix presumably doing all the work of explaining what these challenges really are, how they are evaluated, why they matter, why they were chosen, etc. I think more of that belongs in the paper. I’d rather see a paper arguing for and introducing a benchmark do MUCH more of that motivation and explanation than a slog through the performance on the benchmark of a bunch of models. After all, why do we care how these models perform on problem x before we know what problem x is and why it matters? That said, perhaps domain experts already know these challenges so well that the current level of text is sufficient? I am not in a position to judge, but I think the paper would likely be higher-impact and better if it was more readable by non-insiders. I was hoping to learn a LOT about this area by reading this manuscript, and instead, sadly, I learned basically nothing (except there is a new benchmark). The reason I am still voting for publication despite that is I imagine this paper is quite valuable for insiders, but great papers do better at offering something to non-insiders.
>
> **[A2]** We thank the reviewer for the valuable feedback regarding the paper's structure and content. In our revised version, we provide a  manuscript structure where the main manuscript presents key evaluation results, enabling readers to quickly grasp the overall landscape of tasks and performance of protein foundation models. For researchers interested in detailed task motivations, thoughts on metric selections, dataset considerations, implementation information, and result analysis, we have placed the detailed information in the appendix.
> 1. Specifically, we have reorganized the appendix (Section B) using a task-centric approach. For each task, we now provide comprehensive details including:
>   - Task definitions
>   - Metric justifications and descriptions
>   - Dataset specifications
>   - Supplementary results
>   - Potential discussions to provide more insights
> 2. To facilitate navigation, we added direct links in each task section that connect to their corresponding detailed explanations in the appendix.
> 3. The reorganized information is now highlighted in blue for improved visibility.
>
> Additionally,  we will also release a task-centered arxiv paper. This format allows researchers to access task-specific information efficiently.

---

> ### Author Response · Authors · 2024-11-21
> **Rebuttal by Authors**
>
> **[Q3]** Minor: You say ProteinBench provides a holistic view. That is too strong. It is virtually impossible to provide a holistic view of such a complex topic. Please switch such language to “more holistic” or similar. You say benchmarks are crucial or critical for progress. This is a strong claim. I think they are helpful, but progress can be made without them, and they can actually hurt progress too (see Why Greatness Cannot Be Planned), so I recommend a more nuanced statement.
>
> **[A3]** We appreciate the reviewer’s feedback on the term "holistic benchmark." Our goal is to convey that we aim for a more comprehensive benchmark. As demonstrated in Table 11, our benchmark is the most extensive study in the field, addressing an urgent need by covering multiple tasks. In the discussion section of the manuscript, we acknowledge that the current version is limited by the evaluation of a restricted number of methods. We envision our benchmark as an evolving tool and are committed to its ongoing optimization in the future.

---

> ### Author Response · Authors · 2024-11-21
> **Rebuttal by Authors**
>
> **[Q4]** Line 181: You say this finding has significant implications for the field, but it is drawn from 2 data points only! Please properly soften the claims. Line 192: “We have noticed that….” That’s a weird way to describe your own work/choices. You sound surprised you didn’t include more methods? I suggest finding clearer, less confusing language.
>
> **[A4]** We thank the reviewer for bringing this to our attention. We polish the statement in the revised manuscript by softening the claims into "This finding suggests no single model currently excels across all inverse folding objectives. The choice of model should be carefully aligned with the intended applications."

---

> ### Author Response · Authors · 2024-11-21
> **Rebuttal by Authors**
>
> **[Q5]** Note: I am rating this a 6 and not an 8 simply because I do not know enough to evaluate the vast majority of the work. If the other reviewers think it is good, then this 6 should support publication. But I do not want to set a super high score that would override experts with doubts. If all experts agree the technical work and novelty are solid, I'd be happy with an 8.
>
> **[A5]** Many thanks for the reviewer's feedback. We appreciate all the comments and understand the perspective. We hope that our revisions address the concerns and provide clarity on the technical work and novelty of our study. We look forward to the insights of the other reviewers and hope for a positive evaluation.

---

### Official Review · Reviewer_jXTi · 2024-11-08

**Soundness:** 2
**Presentation:** 3
**Contribution:** 3
**Rating:** 8
**Confidence:** 4

**Summary:**

The authors introduce a standardized benchmarking framework to evaluate the performance of protein foundation models. The framework includes 1) task taxonomy, categorizing key tasks in protein modeling (protein design, confirmation prediction);  2) multi-metric evaluation across quality, novelty, diversity, and robustness dimensions. 3) In-depth analyses based on different user objectives. The study finds that thorough evaluation metrics are crucial for adequately validating protein models and that no single model is optimal across all protein design tasks, which underlines the need to match models to specific applications. The authors intend the framework to serve as a collaboratively developed comprehensive and transparent benchmark for evaluating protein foundation models.

**Strengths:**

-	The framework’s taxonomy of tasks within the domain of protein foundation models is insightful.  It makes it easier to evaluate where each model excels or falls short.
-	The multi-dimensional metrics aims to capture various aspects of model performance which is appropriate given the complexity of the protein modeling.
-	The authors conduct a large number of experiments, demonstrating the breadth of the evaluation and ensuring the results' validity across various models and tasks.
-	Leaderboard and open-source code can potentially facilitate more fair comparison and promote transparency.

**Weaknesses:**

-	Given that the authors have made an extensive amount of experimental study, some reorganization of the paper could strengthen the delivery of the contributions of the paper. Including clear and complete definitions, explanations, and relevance of the metrics would be helpful. The relevance and insights of the results could replace the explanations of the results. For example, Section 2.2.6 Antibody Design, instead of listing the outperforming models for evaluation, which is provided in Table 6, authors could discuss the relevance of these metrics along with the insights gained from the results similar to the one that they provided in the last paragraph.
-	Lack of consistency in the training data across models is a limitation that undercuts the one of the main promises of the proposed framework which is standardization of the evaluation of protein foundation evaluation. This may not be an issue in the future as the framework is further developed and more mature.
-	The authors mention that no model performs optimally across all metrics. It does not, however, fully explore the causes and potential trade-offs. Insights into why certain models perform better for certain tasks would provide better guidance on choosing task-appropriate models.
Minor Issues:
-	The description of Table 12 does not align with the data presented in the table
-	Items (3) and (4) in the conclusion are the same.
-	Figure 2 is too small.

**Questions:**

- The paper highlights four key dimensions (quality, novelty, diversity, and robustness), but the result tables, including Table 1, emphasize only the first three dimensions. Why are robustness metrics omitted from the tables? Additionally, some dimensions in Table 1, such as those in antibody design, do not align with these four key categories.
-	Given that no model excels across all metrics, how should the models be ranked on the leaderboard, given the trade-offs across different metrics?

---

> ### Author Response · Authors · 2024-11-21
> **Rebuttal by Authors**
>
> We greatly appreciate your constructive feedback regarding the manuscript's organization and depth of analysis. In response, we have implemented two major improvements: First, we have thoroughly restructured the manuscript and appendix using a task-centric approach, ensuring clearer organization and logical flow of information. Second, we enhanced our analytical depth by providing comprehensive discussions and deeper insights for each protein design task. We believe these revisions substantially strengthen the manuscript and welcome your assessment of these changes.
>
> **[Q1]** Given that the authors have made an extensive amount of experimental study, some reorganization of the paper could strengthen the delivery of the contributions of the paper. Including clear and complete definitions, explanations, and relevance of the metrics would be helpful.
>
> **[A1]** We thank the reviewer for their constructive feedback on the manuscript's organization. To enhance clarity and accessibility, we have implemented several structural improvements:
> 1. We have reorganized the appendix (Section B) using a task-centric approach. For each task, we now provide comprehensive details including:
>   - Task definitions
>   - Metrics justification and descriptions
>   - Dataset specifications
>   - Supplementary results
>   - Extended discussions to provide more insights
> 2. To facilitate navigation, we have added direct links in each task section that connect to their corresponding detailed explanations in the appendix.
> 3. The reorganized information is now highlighted in blue for improved visibility.
> 4. We can offer a task-centered organization of the manuscript for our leaderboard and arXiv versions, which are not subject to page limits.

---

> ### Author Response · Authors · 2024-11-21
> **Rebuttal by Authors**
>
> **[Q2]** The relevance and insights of the results could replace the explanations of the results. For example, Section 2.2.6 Antibody Design, instead of listing the outperforming models for evaluation, which is provided in Table 6, authors could discuss the relevance of these metrics along with the insights gained from the results similar to the one that they provided in the last paragraph.
>
> **[A2]** We recognize the importance of providing insights for the results in our benchmark. In the revised manuscript, we have expanded the further analysis and insights gained from the results, adding this information in Appendix B for each task in the bulletin titled **[Extended Explanations and Discussion on Model Performance]**. Two examples of inverse folding and antibody design is attached here.
>
> Inverse folding:
> 1. Dataset Characteristics Impact Performance: Our evaluation spans two dataset types: high-quality experimental structures (CASP/CAMEO) and computational de novo structures containing inherent noise. Models performing well on the more challenging de novo structures demonstrate superior robustness, as they must overcome structural uncertainties while maintaining design accuracy.
> 2. Training Strategy Influences Robustness: ProteinMPNN's approach of incorporating backbone noise during training proves highly effective. Our results confirm their findings that increased training noise correlates with improved model robustness. This is evidenced by ProteinMPNN's superior performance in de novo backbone-based sequence design, validating backbone noise augmentation as an effective strategy for enhancing model resilience.
>
> Antibody analysis:
>
> The involved models differ mainly in modeling methods and initialization methods.
>
> HERN stands out as the only autoregressive generative model, excelling in sequence naturalness by effectively capturing amino acid dependencies. Unlike the non-autoregressive methods, like MEAN and dyMEAN, which fail in modeling dependencies between residues (reducing to focus on marginal distributions at each CDR position and thus lost the sequence specificity towards different antigens), HERN’s explicit modeling of inter-residue relationships within CDR-CDR and CDR-FR contributes significantly to sequence rationality;
>
> MEAN achieved the best RMSD among all methods, which we attribute to its unique structural initialization method. Unlike diffusion-based methods that use noise from N(0,I) for structure initialization, MEAN performs a linear structural initialization between FR residues that connect the CDR regions. This initialization method potentially provides a better starting point for structure generation, and also ensures that the residues at both ends of the CDR are not too far from their actual positions;
>   The diffusion-based methods (DiffAb & AbDPO) generally perform better in generating more reasonable structures, with better C-N bond length, less clashes and lower energy, which demonstrate the advantage of diffusion models in structural modeling.

---

> ### Author Response · Authors · 2024-11-21
> **Rebuttal by authors**
>
> **[Q3]** Lack of consistency in the training data across models is a limitation that undercuts the one of the main promises of the proposed framework which is standardization of the evaluation of protein foundation evaluation. This may not be an issue in the future as the framework is further developed and more mature.
>
> **[A3]** We thank the reviewer for bringing this to our attention.
>
> 1. In our manuscript, all models for the antibody design task were retrained using the same dataset. This consistency enables a direct comparison of their underlying technical approaches. However, for the other tasks, the training datasets varied, which may affect the comparability of the results.
> 2. Our current benchmarking approach focuses on evaluating existing methods and models at the model layer rather than the technique layer, where training data is considered an integral part of each method's strategy. This approach aligns with other established benchmarks of foundation models that standardize model evaluation rather than isolating technical components. We believe this better serves users by providing insights into real-world model performance.
> 3. we recognize the importance of controlling for training data differences. We envision ProteinBench as an evolving benchmark, and in future iterations, we plan to implement more rigorous controls to account for these differences in training data. This will help provide deeper insights into the impact of training data on model performance.

---

> ### Author Response · Authors · 2024-11-21
> **Rebuttal by Authors**
>
> **[Q4]** The authors mention that no model performs optimally across all metrics. It does not, however, fully explore the causes and potential trade-offs. Insights into why certain models perform better for certain tasks would provide better guidance on choosing task-appropriate models.
>
> **[A4]** We thank the reviewer for highlighting this concern. In the revised manuscript, we have expanded our analysis and insights based on the results, incorporating this information in Appendix B for each task under the section titled [Extended Explanations and Discussion on Model Performance].
>
> For instance, we discuss the trade-offs between quality and diversity in the backbone design task.
>
> A notable observation across various backbone design methods is the inverse relationship between structural quality and diversity: as methods produce structures with less quality, the diversity and novelty of the generated backbones tend to increase. We emphasize that structural quality should be considered the primary metric, as diversity and novelty are meaningful only when the generated structures maintain sufficient quality. Without adequate structural quality, high diversity or novelty scores may merely indicate the generation of unrealistic or physically implausible conformations.

---

> ### Author Response · Authors · 2024-11-21
> **Rebuttal by Authors**
>
> **[Q5]** Minor Issues:
> - The description of Table 12 does not align with the data presented in the table
> - Items (3) and (4) in the conclusion are the same.
> - Figure 2 is too small.
>
> **[A5]** We thank the reviewer for bringing this concern to our attention. We corrected these typo errors in the revised manuscript.
> 1. We have corrected the description of Table 12.
> 2. We removed the repeated conclusion.
> 3. Figure 2 is enlarged.

---

> ### Author Response · Authors · 2024-11-21
> **Rebuttal by Authors**
>
> **[Q6]** The paper highlights four key dimensions (quality, novelty, diversity, and robustness), but the result tables, including Table 1, emphasize only the first three dimensions. Why are robustness metrics omitted from the tables? Additionally, some dimensions in Table 1, such as those in antibody design, do not align with these four key categories.
>
> **[A6]** We thank the reviewer for the response.
> 1. We updated Table 1 in the revised manuscript by adding robustness.
> 2. Antibody design represents a specialized case requiring task-specific evaluation criteria. We maintain separate evaluation dimensions for antibody design due to its unique therapeutic applications and specialized requirements. These specialized metrics better capture the essential characteristics needed for therapeutic antibody development.

---

> ### Author Response · Authors · 2024-11-21
> **Rebuttal by Authors**
>
> **[Q7]** Given that no model excels across all metrics, how should the models be ranked on the leaderboard, given the trade-offs across different metrics?
>
> **[A7]** We rank the models on the leaderboard using the mean score across all metrics. This approach provides a balanced overview of model performance, acknowledging the trade-offs between different metrics while allowing for a comprehensive comparison.

---

> ### Author Response · Authors · 2024-11-26
> **Looking forward to hearing your feedback!**
>
> Thank you for taking the time to review our paper. We have polished our paper following your suggestions. We hope our responses have addressed your concerns raised so far. In case of any unresolved questions or further concerns, please let us know.
>
> Best wishes,
>
> Authors

---

> ### Comment · Reviewer_jXTi · 2024-11-26
>
> The authors provided appropriate, and detailed responses and addressed this reviewer’s concerns. I'll update my rating appropriately. Thanks

---

> > ### Author Response · Authors · 2024-11-26
> >
> > Thank you for raising your rating. We're glad that our rebuttal has fully addressed your concerns. Your thoughtful comments and feedback have been invaluable in helping us improve the paper.

---

### Author Response · Authors · 2024-11-21
**We are pleased to hear any feedback from all reviewers before discussion deadline**

Dear Reviewers, ACs, and SACs,

We would like to sincerely thank all the reviewers for their efforts and valuable suggestions. We have made every effort to address the reviewers' concerns, including the following:

1. Reorganized the structure of the manuscript.
2. Provided implementation details for the benchmark.
3. Expanded the discussion and insights for each task.
4. Included rational explanations for non-expert readers.
5. Corrected minor presentation issues.

We appreciate everyone's time and effort in providing insightful feedback, which has greatly helped us improve our manuscript. We have revised the paper to incorporate many of the reviewers' suggestions and comments, and we are truly grateful for your contributions.

We welcome any further feedback during the discussion phase!

Thank you once again!

Best regards,
Authors

---

### Meta-Review · Area_Chair_febm · 2024-12-20

**Metareview:**

This paper claims that so far, there have been numerous trained foundation models for protein prediction, but there haven't been standardized benchmarks to evaluate their performance on downstream tasks. This paper proposes a comprehensive list of over 9 benchmarks to assess notions of (Accuracy, Functionality, Specificity, Rationality, Quality, Novelty, Diversity), and also experimented with 20+ official models from the field to list their performances.

## Strengths
* The authors seemed to have spent a great deal of effort in providing a unified benchmark set, and have diligently evaluated many of the most recent and SOTA protein models so far. Appendix B showed that they used the official codebases from different models, which must've taken a large amount of work trying to get every model working.
* From my (admittedly limited understanding), this may be one of the first large-scale efforts to setup an official benchmarking and leaderboard for protein tasks.

## Weaknesses
* For a non-expert in this domain, the paper is definitely hard to learn and read from. It's understandable that as a benchmarking paper, there are naturally going to be lots of scores and results involved, but it's impossible for me to decipher the meanings behind the results on "peptide bond breaking" or "ligand binding"
* It appears that even domain-expert reviewers feel like this paper doesn't do a great job at providing the significance and insights of the results, other than blanket conclusions such as "no model does the best at everything".

**Additional Comments On Reviewer Discussion:**

Due to the subject being quite specific to biology and protein design, I must admit I'm also not an expert in this field, and therefore I needed to very carefully read the reviews of those who are in the field (Reviewers jXTi, d7D6), while I and Reviewers (bamz, hosV) can be considered general machine learning researchers.

Domain expert reviewers:
  * Specific details on dataset curation + splitting are extremely important and required for the benchmarking to be trustworthy. These details need to be laid out explicitly.
  * Certain metrics such as scRMSD in antibody design can be very misleading and not the true objective which should be optimized.
  * Details such as temperature sampling may have been slightly unfair to use.
  * This paper contains benchmarks about both "protein design and protein conformation prediction tasks", while it may have been better to scope it only for protein design tasks.

General ML reviewers:
  * The paper doesn't do a good job of explaining why it's important to outsiders (after all, this is still ICLR and not e.g. Nature).
  * Post-rebuttal, most of the explanations were only added to the Appendix rather than main body, which still doesn't resolve the outsider's readability issue.

Common issues:
  * The paper is written as a "laundry list" of results without explaining the importance and significance of the metrics. Additional insights into why "model X underperformed on benchmark Y" should also be provided.
  * Foundational model Training data should be standardized
    * I + the authors would consider this a moot point, since most models train on different large-scale datasets in general, and indeed, they can be seen as part of the model recipe itself.

Post-rebuttal, most of these issues haven't been truly resolved (judging by the blue text indicating updates and edits), and thus for now, the paper definitely can be accepted as a poster, but I'm not confident about moving to anything higher, e.g. spotlight or oral.

---

### Decision · Program_Chairs · 2025-01-22

Accept (Poster)